# This Looks Like Those: Illuminating Prototypical Concepts Using Multiple Visualizations

**Chiyu Ma**[*]
Dartmouth College
chiyu.ma.gr@dartmouth.edu

**Brandon Zhao**[*]
Caltech
byzhao@caltech.edu

**Chaofan Chen**
University of Maine
chaofan.chen@maine.edu

**Cynthia Rudin**
Duke University
cynthia@cs.duke.edu

## Abstract

We present ProtoConcepts, a method for interpretable image classification combining deep learning and case-based reasoning using prototypical parts. Existing work in prototype-based image classification uses a "this looks like that" reasoning process, which dissects a test image by finding prototypical parts and combining evidence from these prototypes to make a final classification. However, all of the existing prototypical part-based image classifiers provide only one-to-one comparisons, where a single training image patch serves as a prototype to compare with a part of our test image. With these single-image comparisons, it can often be difficult to identify the underlying concept being compared (e.g., "is it comparing the color or the shape?"). Our proposed method modifies the architecture of prototype-based networks to instead learn prototypical concepts which are visualized using multiple image patches. Having multiple visualizations of the same prototype allows us to more easily identify the concept captured by that prototype (e.g., "the test image and the related training patches are all the same shade of blue"), and allows our model to create richer, more interpretable visual explanations. Our experiments show that our "this looks like those" reasoning process can be applied as a modification to a wide range of existing prototypical image classification networks while achieving comparable accuracy on benchmark datasets.

## 1   Introduction

As machine learning models are increasingly adopted in high-impact, high-stakes domains such as healthcare [2, 13], self-driving cars [24], facial recognition [23], etc., the design of human-interpretable models has become essential for ensuring accountability and transparency of their decisions. In particular, a class of models called prototype networks have combined the power of deep learning with the explainability of case-based reasoning to make accurate, human-interpretable decisions on fine-grained image classification tasks [3, 6, 19, 25, 26, 36]. These models make decisions by dissecting an image into informative feature patches, then comparing them to prototypical parts learned during training. Evidence of similarity to each prototypical part is then combined to make a final classification.

When evaluating the interpretability of an image classification network, it is important to consider not only the model's capability to explain its reasoning process, but also the quality of its explanations. The image features analyzed by prototype networks can encode a variety of visual properties, such as color, shape, texture, contrast, or saturation. Because each prototype in the existing prototype network corresponds exactly to a single patch of a single training image, it can be difficult for human users to understand the exact semantic content underlying a prototype's visualization. Although there has been some work in quantifying the importance of each of these feature categories in prototypes [20], the exact semantic content in an image patch can still remain ambiguous (e.g. "Texture is

---

[*]Denotes equal contribution

37th Conference on Neural Information Processing Systems (NeurIPS 2023).

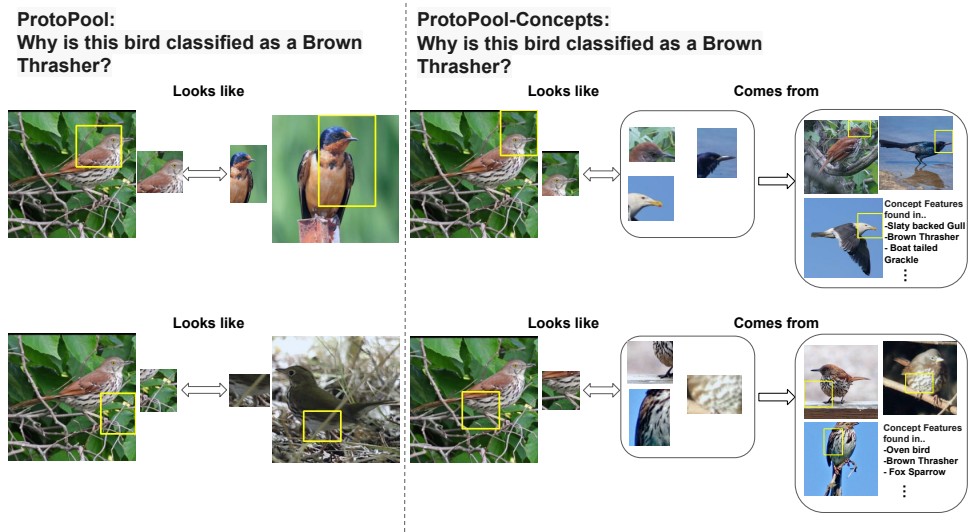

Figure 1: Image of a Brown Thrasher and how the ProtoPool (left) and ProtoPool-Concepts (ours, right) explain their decisions. Prototype classifications are made by finding patches in the image similar to learned prototypical parts. Single-visualization methods such as ProtoPool can make visually ambiguous decisions when the semantic features underlying a prototype are unclear. Our method provides clearer explanations by providing multiple visualizations of prototypical concepts found in the test image.

important in this prototype, but what kind of texture in particular?"). The ambiguity underlying prototype visualizations has been shown to cause a gap between visual explanations from prototype networks and human understanding of visual similarity [15]. Additionally, a variety of prototype networks have been proposed in which prototypical parts from a single image can serve as evidence for many different classes, thus reducing the total number of prototypes needed for the network to make classifications [19, 26, 25]. The rationale behind this prototype sharing is that certain distinctive features (e.g. "long beak, solid black head, spotted belly") can be found in multiple classes, and thus do not need to be represented as distinct class-specific copies. However, while reducing the number of prototypes can simplify visual explanations by reducing the amount of considered evidence, these methods can often produce seemingly nonsensical explanations when a training patch from an image is used spuriously as evidence for a different class.

Inspired by the difficulty of identifying the meaning behind single-image prototypes, we propose ProtoConcepts, a novel modification to prototype geometry enabling visualizations of prototypical concepts from multiple training images. This allows human users to better understand the conceptual content of a prototype by observing the features shared among its visualizations. An example of a visual explanation produced by our ProtoConcepts is shown in Figure 1 (right). Unlike previous prototype networks where a prototype is visualized using a single training image patch (Figure 1, left), our ProtoConcepts visualizes each prototype with multiple visualizations. Because our model offers many visual examples for each prototype, the semantic meaning of each prototype can be determined with less ambiguity – we can look at the commonality between the visualized training image patches to infer the semantic meaning of each prototype. Our method can be applied to a wide range of prototype-based networks, resulting in better human interpretability while achieving comparable classification accuracy. Our code is available at https://github.com/Henrymachiyu/This-looks-like-those_ProtoConcepts

## 2   Related Work

Attempting to understand deep neural networks, *posthoc* explanation techniques such as activation maximization [7, 21, 37, 38], image perturbation [8, 12], and saliency visualizations [30–32, 27] fail to explain a network's reasoning process, and their results can be risky and unreliable [18, 1]. On the other hand, prototype-based networks compare learned latent feature representations, called

prototypes, to the latent representations of an image to perform classification. These models are constrained to be intrinsically interpretable. The Prototypical Part Network (ProtoPNet) [3] uses class-specific prototypes, where a pre-determined number of prototypes are assigned to each class. Each prototype is trained to be similar to feature patches from images of its own class and dissimilar to patches from images of other classes. This gives such class-specific models high discriminative power suitable for fine-grained image classification. In the end, prototype similarity scores are aggregated as positive evidence for each class in a "scoresheet." The Transparent Embedding Space Network (TesNet) [36] improved on the class-specific ProtoPNet architecture by transforming the latent space geometry into a hypersphere, and representing prototypes as near-orthogonal basis vectors specific to a class subspace in the Grassmannian manifold.

Other prototype-based networks found that abandoning class-specific prototype constraints allowed for a significant reduction in the number of prototypes, making explanations simpler and allowing prototypes to represent similar visual concepts shared in images of different classes. In [25], a ProtoPNet is first trained with enforced class specificity, and then a novel data-based similarity metric is used to "merge" highly similar prototypes across classes into a single prototype representing multiple classes. In Neural Prototype Trees (ProtoTree) [19], the "scoresheet" of the ProtoPNet is replaced by a binary decision tree, in which the scored presence or absence of a prototype in each node determines the traversal path of the tree. Thus, each prototype in the tree represents either positive or negative evidence for the classes at the leaf nodes of the subtree for which the prototype is in the root node. Finally, in the ProtoPool network [26], the soft assignment of prototypes to classes is learned throughout training, and is eventually transformed into a hard assignment at test time.

Our work also relates to works that try to understand the semantic meaning behind a learned prototype of a prototype-based network. For example, in [20], Nauta et al. developed a method to quantify the importance of various visual characteristics relevant to each prototype. Our approach is complementary to, yet differs from theirs in that we illuminate the concept behind each prototype by presenting multiple visualizations of the same prototype, thereby allowing one to better infer the semantic meaning behind each prototype.

Finally, our work is related to posthoc methods in concept visualization such as TCAV [14] and ACE [9]. *Posthoc* methods do not show any part of the model's actual reasoning process. Worse, their visual explanations may not be faithful to the model's reasoning process. Sometimes, multiple concepts have the same concept vector [as discussed in 5], and concepts may not be clustered in the latent space (since they are not trained to be that way), leading to meaningless concept vectors. Instead, our method is inherently interpretable, and derives its visual concept-based explanations directly from the decision-making process.

## 3  Methods

We begin with a brief general review of traditional prototype-based models, then propose our method, which represents prototypes as interpretable concepts with multiple visual explanations. We show how our method can be applied as a modification to existing prototype-based networks, and introduce a novel training algorithm to learn meaningful prototypical concepts. Details for the implementation of specific prototype-based classification models are in the Experiments section.

### 3.1  Prototype Architectures

Existing prototype-based networks [3, 19, 25, 26, 36] can be understood architecturally as the combination of three components. First is a convolutional neural network (CNN) $f$ parameterized by weights $w_{\text{conv}}$, which acts as a latent feature extractor on input images $\mathbf{x}$. Next the prototype layer $p$ considers the convolutional output $f(\mathbf{x})$ as H × W "patch" vectors $z_i \in \mathbb{R}^D$, each of which corresponds to features from a patch of the original image space. Each patch is compared to $m$ learned prototype vectors $\mathbf{P} = \{\mathbf{p}_j\}_{j=1}^m$, where each prototype $\mathbf{p}_j \in \mathbb{R}^D$ lies in the same space as the image's latent feature patches. The prototype layer $p$ calculates a similarity score $g_{\mathbf{p}_j} \circ f(\mathbf{x})$ for each prototype $\mathbf{p}_j$ which is monotonically decreasing with respect to the distance between $\mathbf{p}_j$ and the closest image patch $\tilde{\mathbf{z}} \in f(\mathbf{x})$ in the model's feature space. Finally, the prototype layer $g_{\mathbf{p}}$ is followed by a prototype class assignment mechanism $h$ which assigns evidence logits to each class based on the calculated prototype similarity values $g_{\mathbf{p}_j} \circ f(\mathbf{x})$ and class assignment weights $w_h$. The evidence

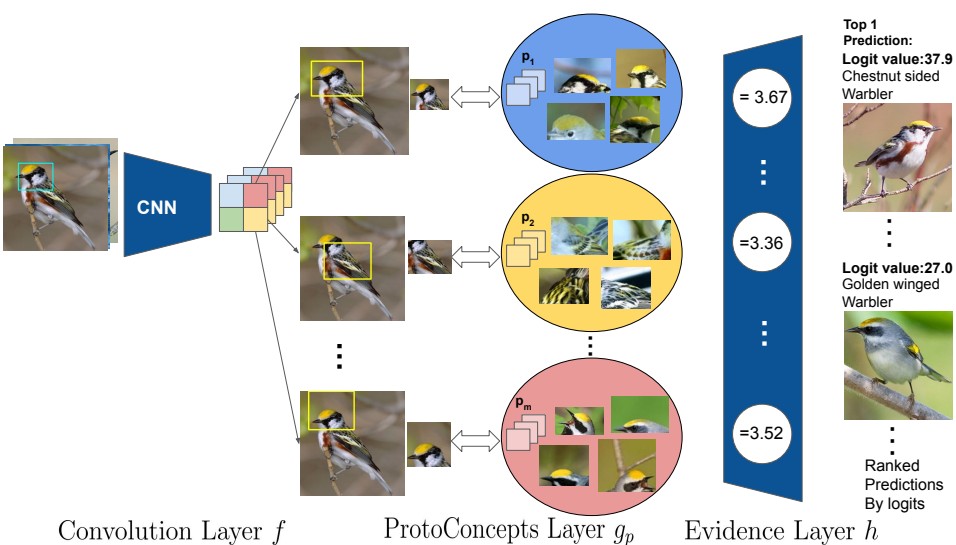

Figure 2: Architecture of ProtoConcepts. The convolutional layer includes all types of architectures, including ResNet, VGG and DenseNet. Evidence Layer $h$ consists of a prototype class assignment mechanism and a fully connected layer. Depending on the specific prototype-based architecture, TesNet and ProtoPNet would have a one-to-one prototype class assignment mechanism, while ProtoPool and ProtoPShare would have a shareable prototype mechanism among the classes.

logits are then normalized by a softmax to yield predicted probabilities for a given image belonging to each class.

In the final model, each prototype vector $\mathbf{p}_j$ is restricted so that it is equal to a latent patch $\tilde{\mathbf{z}} \in f(\mathbf{x}_i)$ from some training image $\mathbf{x}_i$. Thus, the model's reasoning process can be interpreted visually as an evidence scoresheet of comparisons between test image patches and salient features underlying previously-seen training image patches. We propose to modify the geometry of prototypes $\mathbf{p}_j$ by instead representing them as a *ball* in the latent space encompassing *multiple* training image patches. Mathematically, each prototype is a ball $\mathbf{p}_j = B(\mathbf{c}_j, r_j)$ around a central vector $\mathbf{c}_j \in \mathbb{R}^D$ with radius $r_j \in \mathbb{R}^+$. This ball representation has the property that the similarity scores of any latent patch vector $\mathbf{z}_i \in \mathbb{R}^D$ and any vector $\mathbf{d}_j \in \mathbf{p}_j$ in the prototypical ball are constrained to be the same. Hence, with this design, as long as a prototypical ball $\mathbf{p}_j$ contains a number of latent training patches, we can visualize the prototype $\mathbf{p}_j$ by visualizing any of those latent training patches it contains. This set representation allows our prototypes to represent concepts found in all latent training patches contained within a prototypical ball, thereby providing richer semantic explanations than that of visualizing just a single latent training patch. Implementing our concept geometry on a prototype-based model entails minor changes to the architecture and loss; model-specific implementation details are described in Sec. 4.1.1.

## 3.2 Prototypical Concept Representations

The prototype layer $g_{\mathbf{p}_j}$ of previous prototype-based classification models uses one of two activation functions for calculating similarity scores between prototypes and image patches: a log-based activation on inverse $\ell^2$ distance, and a cosine similarity score. In this section, we formalize our ball representation of prototypical concepts with respect to the two corresponding latent spaces.

**Log-based Activation:** Models such as ProtoPNet [3] and ProtoPool [26] use a log-based function $g^{\log}$ to compute a similarity score, which is monotonically decreasing with respect to the $\ell^2$ distance. In particular, for the latent representation $\mathbf{z} = f(\mathbf{x})$ of an input image $\mathbf{x}$, this function computes $g_{\mathbf{p}_j}^{\log}(\mathbf{z}) = \max_{\tilde{\mathbf{z}} \in \text{patches}(\mathbf{z})} \log((\|\tilde{\mathbf{z}} - p_j\|_2^2 + 1)/(\|\tilde{\mathbf{z}} - p_j\|_2^2 + \epsilon))$ with a small constant $\epsilon$ to avoid numerical overflow. For prototypical concepts, we propose an analogous similarity function

corresponding to the distance to center $\mathbf{c}_j$ of prototypical ball $\mathbf{p}_j = B(\mathbf{c}_j, r_j)$, bounded below by $r_j$:

$$\tilde{g}_{\mathbf{p}_j}^{\log}(\mathbf{z}) = \max_{\tilde{\mathbf{z}} \in \text{patches}(\mathbf{z})} \min \left( \log \left( \frac{\|\tilde{\mathbf{z}} - \mathbf{c}_j\|_2^2 + 1}{\|\tilde{\mathbf{z}} - \mathbf{c}_j\|_2^2 + \epsilon} \right), \log \left( \frac{r_j + 1}{r_j + \epsilon} \right) \right). \tag{1}$$

**Cosine Similarity:**   Models such as TesNet [36] and Deformable ProtoPNet [6] have found that representing prototypes as unit vectors on a hypersphere can improve the accuracy of prototype-based models. In these models, the prototype layer compares prototypes to latent image patches via the cosine similarity function $g_{\mathbf{p}_j}^{\cos}(\mathbf{z}) = \max_{\tilde{\mathbf{z}} \in \text{patches}(\mathbf{z})} \tilde{\mathbf{z}}^T \mathbf{p}_j / (\|\tilde{\mathbf{z}}\|_2 \|\mathbf{p}_j\|_2)$. This cosine similarity can be interpreted as a cosine activation function on the angular distance between normalized vectors on the hypersphere. For models with a hypersphere-based latent space, we replace the cosine similarity function $g_{\mathbf{p}_j}^{\cos}$ by calculating the cosine similarity to the ball $B(\mathbf{c}_j, r_j)$ on the hypersphere :

$$\tilde{g}_{\mathbf{p}_j}^{\cos}(\mathbf{z}) = \max_{\tilde{\mathbf{z}} \in \text{patches}(\mathbf{z})} \min \left( \frac{\tilde{\mathbf{z}}^T \mathbf{c}_j}{\|\tilde{\mathbf{z}}\|_2 \|\mathbf{c}_j\|_2}, \cos(r_j) \right). \tag{2}$$

Both Eqs. (1) and (2) ensure that for any patch vector $\mathbf{z}_i \in \mathbb{R}^D$, if $\mathbf{z}_i$ lies outside the ball $\mathbf{p}_j$, its distance to $\mathbf{p}_j$ is the distance to its center $\mathbf{c}_j$, and if $\mathbf{z}_i$ lies inside $\mathbf{p}_j$, its distance to $\mathbf{p}_j$ is kept the same as the radius of the ball. During training, a pass-through estimator of each activation function is used to avoid a zero gradient when training patches lie within the prototypical ball.

### 3.3   Training Algorithm

The training of prototype-based networks is typically divided into three stages: (1) stochastic gradient descent (SGD) of layers before the last layer; (2) projection of prototypes; (3) optimization of the last layer $h$. Because of the ball representation of prototypical concepts, the second stage can be skipped as prototypical balls are visualized by the training patches they contain at the end of the first step.

**SGD of layers before last layer:**   The first training stage prototype-based networks aims to learn a meaningful latent space that clusters important training patches near semantically similar prototypes. To do this, a joint optimization problem is solved to optimize the network parameters using SGD while keeping the last layer weight matrix fixed. To learn semantically meaningful prototypical concepts, we introduce two novel loss terms, $\mathcal{L}_{\text{Clstk}}$ and $\mathcal{L}_{\text{Rad}}$:

$$\mathcal{L}_{\text{Clstk}} = \frac{1}{n} \sum_{i=1}^{n} \sum_{j=1}^{k} \min_{\substack{\mathbf{p}_j \in \mathbf{P}_{y_i} \\ \mathbf{p}_j \neq \mathbf{p}_1, \dots, \mathbf{p}_{j-1}}} \min_{\tilde{\mathbf{z}} \in \text{patches}(f(\mathbf{x}_i))} d(\tilde{\mathbf{z}}, \mathbf{p}_j), \ \mathcal{L}_{\text{Rad}} = \sum_{\substack{\mathbf{p}_j \in \mathbf{P} \\ \mathbf{p}_j = B(\mathbf{c}_j, r_j)}} r_j^2. \tag{3}$$

Here $d$ is used as the distance function corresponding to the respective latent space geometry of the ProtoConcepts module. The minimization of $\mathcal{L}_{\text{Clstk}}$ encourages $k$ prototypical concepts of the correct class to be close to at least one latent patch of each training image. The specific value of $k$ is treated as a hyperparameter and is chosen by cross-validation. As opposed to the traditional prototype cluster loss [3], which only penalizes the distance of the closest prototype to a latent training patch, we find that our top-$k$ formulation is essential for ensuring that prototypical concept balls contain multiple training patches. The minimization of $\mathcal{L}_{\text{Rad}}$ encourages prototypical concept balls to be compact. This allows learned prototypical concepts to encapsulate more specific semantic features important for classification.

**Prototypical Concept Visualisation:**   In previous prototype-based networks, the trained prototype vector $p_j$ is projected (pushed) to the nearest training patch to produce a single patch visualization. However, since we use a set representation for prototypical concepts, it is no longer reasonable to project the prototypical ball or its central vector to the training patches. Instead, we visualize each prototypical concept by the training patches that are contained within each ball at the end of training. Thus, we can produce multiple visualizations from each prototypical concept and skip the projection step altogether.

**Pruning and Fine-tuning:**   Because the number of prototypical concepts is fixed at the beginning of training, it is possible for a trained ProtoConcepts model to have a small number of prototypical

concepts that do not contain any latent training image patches. In other words, these prototypical concepts do not capture any visualizable latent features, and should not be considered when making predictions. To prune the non-visualized prototypical concepts, we design a 0/1 mask on the prototype class assignment mechanism so that the given non-visualized prototypical concepts do not provide evidence for any image classes. After pruning these prototypical concepts, we finally perform a sparse convex optimization on the last layer weight matrix $w_h$ as in previous prototype-based methods [3].

## 4 Experiments

### 4.1 Case Study 1: Bird Species Identification

To demonstrate and examine the effectiveness of our method, we implement the ProtoConcepts module on the ProtoPNet, ProtoPool, and TesNet networks for the Caltech-UCSD Birds-200-2011 (CUB 200-2011) dataset [35]. The dataset contains 5,994/5,794 images for training and testing across 200 different bird species. We perform offline data augmentation, and crop training and testing images using bounding boxes provided with the dataset.

#### 4.1.1 Implementation

In this section, we describe the implementation changes made to each prototype-based model for our experiments to incorporate prototypical concepts. A full list of exact parameter settings and training schedule for each model can be found in the Appendix.

**ProtoPNet:** The ProtoPNet model [3] consists of a CNN backbone, a log-based prototype layer $g^{\log}$, and a fully connected layer $h$ [3]. Each of 200 classes is assigned 10 class-specific prototypes, which are constrained to provide evidence only for their corresponding class through the last layer weighting. We train a ProtoPNet-Concepts model by substituting the prototype layer as in Eq. (1), and then training with additional losses $\mathcal{L}_{\text{Rad}}$ and $\mathcal{L}_{\text{Clstk}}$ with $k = 10$, radius initialization 7.5, and loss weights 0.01 and 0.8, respectively. The training schedule and other hyperparameters are unchanged from the original paper.

**TesNet:** The TesNet model [36] uses a cosine similarity function in its prototype layer to represent the latent space as a hypersphere and incorporates several hypersphere-based loss functions during training. After modifying the activation function as in Eq. (2), we train TesNet-Concepts with a radius initialization of 8.05, and a top-$k$ cluster loss $\mathcal{L}_{\text{Clstk}}$ with $k = 3$. We set the weight for the radius loss as $3 \times 10^{-5}$ and the weight for top-$k$ cluster loss as 0.8 with a learning rate of $1 \times 10^{-4}$ during the warmup stage, and reweight the loss functions described in its original paper with a new training schedule.

**ProtoPool:** The ProtoPool architecture [26] employs a log-based activation on the prototype layer, and adds a prototype class assignment mechanism that allows sharing prototypes among multiple classes, thereby reducing the number of prototypes. After modifying the prototype layer as in Eq. (1), we train ProtoPool-Concepts using a radius initialization of 4.5 and top-$k$ cluster loss $\mathcal{L}_{\text{Clstk}}$ with $k = 10$. We set a radius loss weight $3 \times 10^{-3}$, a top-$k$ cluster weight 0.8, and learning rate for the radius as $0.5 \times 10^{-3}$. The training schedule and other hyperparameters are left unchanged from the original paper.

For baseline comparisons, we retrain CNN backbones and previous prototype-based models using publicly available code. As shown in Table 1, for the same CNN architecture, our ProtoConcepts network achieves similar test accuracy to corresponding baseline models.

**Radius Study:** We performed an ablation on ProtoPool-Concepts with a different set of radius initialization and top-$k$ cluster loss with $k = 10$ under the same training settings described above. Intuitively, a larger radius would encourage more prototypical concepts to capture information and an infinitesimal radius would be equivalent to a single visualization. As shown in Table 2, the number of visualizable prototypical concepts increases with the radius. In addition, radius also influences the accuracy of our ProtoConcepts model after pruning and fine-tuning. If the radius is too small, the number of visualizable prototypical concepts also tends to be small – this means that pruning will remove too many prototypes, and consequently adversely impact performance. On the other hand, if

Table 1: Comparison of ProtoConcepts implementation on ProtoPNet, ProtoPool, TesNet with the corresponding baselines and various backbone architectures. All algorithms on the same architecture perform similarly; the advantage is instead in interpretability.

| Arch. | Model | Prototype $p$ # | Acc.[%] |
|---|---|---|---|
| VGG19 [29] | **ProtoPNet-Concepts (ours)** | $1993 \pm 3$ | $77.9 \pm 0.2$ |
| | ProtoPNet [3] | 2000 | $78.0 \pm 0.2$ |
| | **TesNet-Concepts (ours)** | $1862 \pm 20$ | $78.2 \pm 0.8$ |
| | TesNet [36] | 2000 | $77.9 \pm 0.1$ |
| | Baseline (given in [3]) | N/A | $75.1 \pm 0.4$ |
| Densenet-161 [11] | **ProtoPNet-Concepts (ours)** | $1980 \pm 2$ | $80.7 \pm 0.4$ |
| | ProtoPNet [3] | 2000 | $80.1 \pm 0.3$ |
| | **TesNet-Concepts (ours)** | $1994 \pm 2$ | $81.4 \pm 0.3$ |
| | TesNet [36] | 2000 | $81.5 \pm 0.3$ |
| | **ProtoPool-Concepts (ours)** | $199 \pm 1$ | $81.5 \pm 0.6$ |
| | Protopool [26] | 202 | $80.3 \pm 0.3$ |
| | Baseline (given in [3]) | N/A | $82.2 \pm 0.2$ |
| Resnet-34 [10] | **TesNet-Concepts (ours)** | $1957 \pm 10$ | $79.8 \pm 0.4$ |
| | TesNet [36] | 2000 | $80.7 \pm 0.3$ |
| | **ProtoPool-Concepts (ours)** | $185 \pm 4$ | $80.4 \pm 0.1$ |
| | ProtoPool [26] | 202 | $80.3 \pm 0.2$ |
| Resnet-50 (iNat) [10] | **ProtoPool-Concepts (ours)** | $188 \pm 2$ | $85.2 \pm 0.2$ |
| | ProtoPool [26] | 202 | $85.5 \pm 0.1$ |
| | ProtoTree [19] | 202 | $82.2 \pm 0.7$ |
| Resnet-152 [10] | **ProtoPNet-Concepts (ours)** | $1972 \pm 10$ | $78.0 \pm 0.1$ |
| | ProtoPNet [3] | 2000 | $78.0 \pm 0.3$ |
| | Baseline (given in [3]) | N/A | $81.5 \pm 0.4$ |

the radius is too large, each prototypical ball would capture too many latent representations whose semantic meanings may be too diverse, thereby making the prototypes less meaningful. Hence, we need to tune the radius to be within a reasonable range.

### 4.1.2 Interpretability

The reasoning process of a ProtoConcepts network mirrors that of a traditional prototype network, but provides multiple visualizations of the prototypes used in its explanations. Figure 3 shows how our model classifies a testing image of a Baltimore Oriole into the correct class. It is worth noting that because of the shareable prototypes mechanism inherited from ProtoPool, the visualizations of a learned prototypical concept now come from different classes. The most representative features that an ornithologist would use to classify an adult male Baltimore Oriole are the bright orange and black plumage and the slender beak [28]. As shown in Figure 3, our model is able to compare the test image patch containing orange and black plumage with the learned prototypical concept that captures similar features from its own class. Additionally, our model compares the patch containing the wings to the prototypical concept shared among Baltimore Orioles and other similar birds such as Orchard Orioles. Lastly, our model compares the part of image that contains a slender beak along with its black head to the prototypical concept that captures the slender beak and black head of birds such as Orchard Orioles and Eastern Towhees.

Figure 1 shows an example of how a test image of a Brown Thrasher is correctly classified by ProtoPool and ProtoPool-Concepts. Because of the prototype sharing mechanism, it is frequently the case that training image patches from birds of one class are used as evidence of a test image belonging to another class. As a result, the similarities presented by ProtoPool's visual explanations can be unintuitive. As shown in the plot, it is unclear how ProtoPool found the head of a Brown Thrasher to be similar to the blue head of a Barn Swallow. Moreover, it is hard to determine whether the model is comparing the background or the belly of the Brown Thrasher, as shown at the bottom. However, comparisons made by our method are much clearer thanks to multi-visualizations. Although both models attempt to compare the lower part of the bird to the learned prototypes, it is clear that our ProtoPool-Concepts is looking at the pattern on the belly, which is an identifying feature of the Brown Thrasher [28].

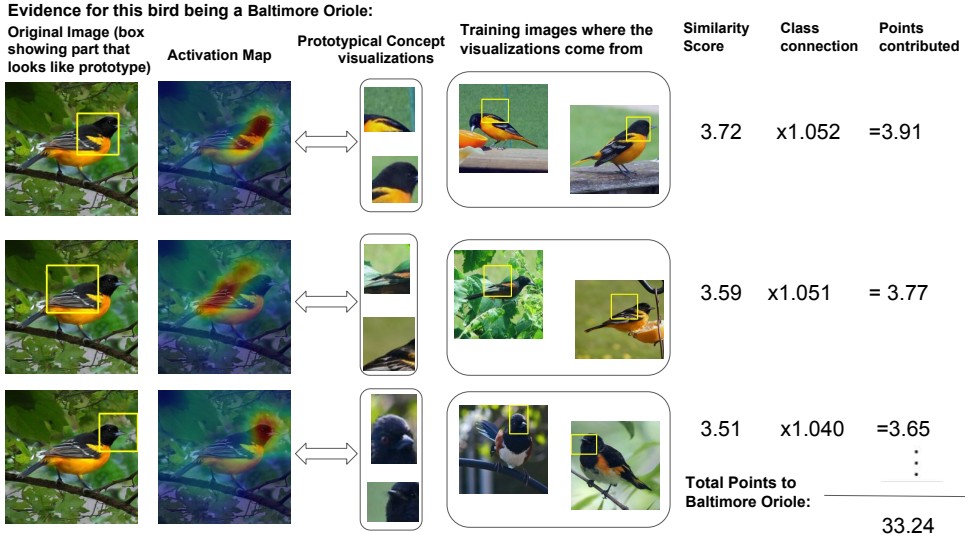

Figure 3: The reasoning process of our network in classifying a test image as a Baltimore Oriole. Test image patches are compared to prototypical concepts in latent space, then visual similarity scores are compiled as evidence for each class.

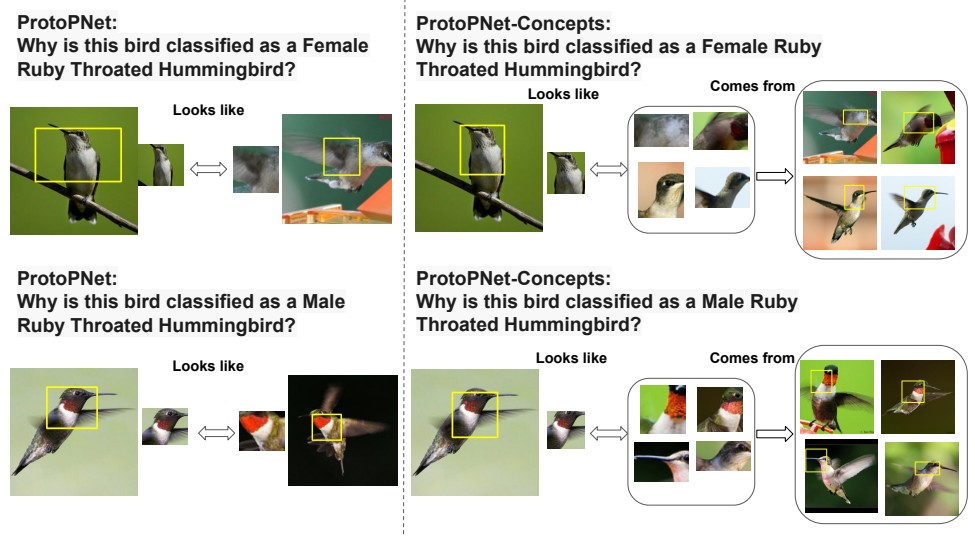

Figure 4: Reasoning of a correctly classified Female Ruby Throated Hummingbird (top) and Male Ruby Throated Hummingbird (bottom) by ProtoPNet (left) and ProtoPNet-Concepts (right).

Figure 4 shows examples of how the two models correctly classify a female and a male ruby-throated hummingbird. In the top row, both of the models found the neck of a female ruby-throated hummingbird from the given test image similar to the white feathers around neck from the training examples. On the other hand, although having a red throat differentiates the male ruby-throated hummingbird from the female [28], the significance of a red-throat to correctly classify a male ruby-throated hummingbird is unclear. As shown in the bottom left of Figure 4, the learned prototype from ProtoPNet is able to compare the prototype that contains a red throat to that of the test image. However, the visualizations from ProtoPNet-Concepts demonstrate that a red throat may not be all that the prototypical concept is representing. It appears to represent broader information about the bird's gray head and the transition to its body, where the additional red or gray color is located on the throat, and the white on the breast.

Table 2: Performance of ProtoPool-Concepts with different radius initializations using ResNet-50(iNat) backbone

| Radius | Acc.[%] before pruning | Acc.[%] after pruning & finetuning | Prototype $p$ # |
|---|---|---|---|
| 4.0 | $85.4 \pm 0.1$ | $81.5 \pm 1.0$ | $61 \pm 6$ |
| 4.5 | $85.2 \pm 0.1$ | $85.2 \pm 0.2$ | $188 \pm 2$ |
| 5.0 | $85.2 \pm 0.3$ | $85.0 \pm 0.3$ | 202 |

Table 3: Accuracy comparison of ProtoPool-Concepts to other shared prototype models on the Stanford Cars dataset

| Arch. | Model | Prototype $p$ # | Acc.[%] |
|---|---|---|---|
| | **ProtoConcepts (ours)** | $150 \pm 6$ | $89.4 \pm 0.5$ |
| Resnet 34 [10] | ProtoPShare [25] | 480 | 86.4 |
| | Protopool [26] | 195 | $89.3 \pm 0.1$ |
| Densenet121 [11] | **ProtoConcepts (ours)** | $194 \pm 1$ | $87.1 \pm 0.4$ |
| | Protopool [26] | 195 | $86.4 \pm 0.1$ |

## 4.2 Case Study 2: Car Model Identification

In this case study, we implement our method for the Stanford Cars dataset [17] of 196 car types, using similar procedures and training algorithms as in the CUB-2011-200 experiment. We trained a ProtoPool-Concepts model with radius initialization $4.5$, keeping other parameter settings the same as described in Sec. 4.1.1. We compare our method to self-reported accuracies of other shared prototype models ProtoPool [26] and ProtoPShare [25]. Our model achieves similar test accuracy to other shared prototype models, with the advantage of multiple visualizations for prototypical concepts. More details, including visual explanations, can be found in the Appendix.

## 5 User Study

To show the reduction of ambiguity and resulting improvement in user interpretability, we created a distinction user study similar to HIVE [15] to compare our ProtoConcepts method with ProtoPNet; results are shown in Table 4. We randomly picked ten samples from the test set and calculated the top two predicted classes (i.e., the classes with the highest predicted probabilities according to the model) for each test sample. We then provided visual explanations from the most activated prototypes for these classes by ProtoPNet and ProtoPNet-Concepts. A test-taker was then asked to choose which class the model is actually predicting, looking only at the visual explanations without the class probabilities. Examples of our user study are shown in Figure 7. We released our user study on Amazon Mechanical Turk and collected 50 responses from test takers with a $98\%$ survey approval rate to ensure the quality of responses, and removed 1 response from both surveys after screening for nonsensical free response answers. We first ran a two-sided $t$-test on self-rated ML experience for the test takers from the ProtoPNet and ProtoPNet-Concept. The $p$-value is 1, and we are assured that there is no statistically significant difference in machine learning experience between the two groups on average. From the results of our survey, we are able to conclude that users with visual explanations from ProtoConcepts outperform those with visual explanations from ProtoPNet to a statistically significant degree ($p = 0.003$). Moreover, users given visual explanations from ProtoPNet were unable to outperform $50\%$ random guessing to a statistically significant degree ($p = 0.289$), whereas users given visual explanations from our model did ($p = 2.85 \times 10^{-5}$). Our survey results show not only that our model provides a notable improvement in user interpretability, but is able to improve non-expert user performance in a difficult fine-grained classification task whereas the previous ProtoPNet model cannot.

## 6 Limitations

The interpretability of prototype-based models comes from *visual* explanations of classification decisions. Hence, the exact semantics (e.g., "this bird has a long beak" or "this bird has a spotted belly") underlying these visual explanations depend on the user's visual system to determine. We present a method for clarifying the user's semantic inference by offering multiple visualizations whose

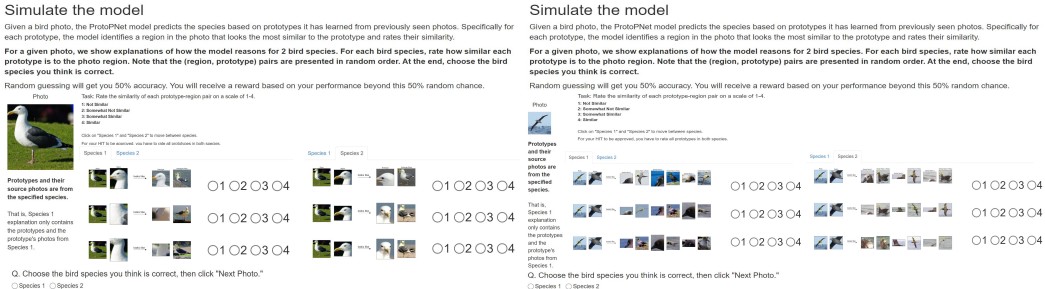

Figure 5: User study example questions with architecture ProtoPNet (left), and ProtoConcepts (right). We showed each user 2 possible class explanations for a given test image and asked the user to pick the correct class. We compared the results from two separate surveys, one in which visual explanations are generated by ProtoPNet, and one in which they are generated by ProtoConcepts.

commonalities can disambiguate our model's exact reasoning. While we find that our concept-based representation is helpful for this process, it is still unable to provide explicit semantics to explain its classifications. With recent progress in large vision-language hybrid models [22, 16, 4, 34], a technique offering explicit semantics for visual explanations could be explored in future work. It is important to note that in some visual domains, concepts may not have natural textual descriptions (e.g., in mammography or other radiology applications where it might look at a patch of a particular type of breast tissue that is difficult to describe and has no associated terminology).

## 7 Broader Impact

Interpretability is an essential ingredient to trustworthy AI systems [33]. Our work presents a major improvement in interpretability to prototype-based networks, which are one of the leading techniques for interpretable neural networks in computer vision. Thus, our technique can be used for important computer vision applications to discover new knowledge and create better human-AI interfaces for difficult, high-impact applications.

Table 4: Results of the user study comparing ProtoPNet to ProtoConcepts. We report the mean user accuracy for each model with a range of plus/minus 1.96 standard deviations. In addition, we conduct two 1-sided t-tests on the results, with the alternative hypotheses that (1) users with ProtoConcepts' visual explanations score higher on average than with ProtoPNet and (2) Each model, respectively, achieves a higher accuracy than random guessing. We conclude that users presented with visual explanations from ProtoConcepts outperform those with explanations from ProtoPNet to a statistically significant degree.

| Model | Mean Acc.[%]±1.96 Std. | p-value (1) | p-value (2) |
|---|---|---|---|
| **ProtoConcepts (Ours)** | **62.1 ± 5.4** | 0.003 | $2.85 \times 10^{-5}$ |
| ProtoPNet [3] | 51.5 ± 5.2 | | 0.288 |

## 8 Conclusion

In this work, we present an interpretable method for image classification which incorporates proto-typical concepts with multiple visualizations to explain its predictions (*this* looks like *those*). Unlike previous works in prototype-based classification which offer only single prototype visualizations, our method allows the user to use commonalities between visualizations of prototypical concepts to better infer the semantic meaning of prototypical concepts. We compare the visual explanations offered by our and previous prototype-based methods and show that our model can achieve comparable accuracy to previous methods.

## Acknowledgements

We gratefully acknowledge support from the National Science Foundation under grants IIS-2130250, RII Track-2 FEC-2218063.

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

# A Ablation Study on Top-$k$ Cluster Loss

To further demonstrate how top-$k$ cluster loss helps the model performance, we performed an ablation study on ProtoPool-Concepts. We trained ProtoPool-Concepts by adding $\mathcal{L}_{\text{Clstk}}$ losses with the choices of $k = 1, 3, 5, 10$ into the model with radius initialization 4.5 using ResNet50 (iNat) backbone. As shown in Table 5, although the classification performance of the model before pruning does not vary much when $\mathcal{L}_{\text{Clstk}}$ is introduced, the number of visualizable prototypes sharply increases, and the performance after pruning is significantly improved. While, as discussed in the main paper, the $\mathcal{L}_{\text{Clstk}}$ term ensures the quality of the latent space, encouraging prototypical concepts which enrich the interpretability of the model.

Table 5: Model Performance on ProtoPool-Concepts under ResNet-50 (iNat) Backbone with different choices of losses. Acc.[%] bf. prune refers to the performance of a model before the pruning procedure and Acc.[%] finetuned refers to the performance after finetuning.

| Model | Prototype $p$ # | Acc.[%] bf. prune | Acc.[%] finetuned |
|---|---|---|---|
| **ProtoPool-Concepts +$\mathcal{L}_{\text{Clstk, k=1}}$** | $50 \pm 1$ | $85.2 \pm 0.3$ | $67.7 \pm 2.0$ |
| **ProtoPool-Concepts + $\mathcal{L}_{\text{Clstk, k=3}}$** | $85 \pm 8$ | $85.3 \pm 0.3$ | $83.5 \pm 0.5$ |
| **ProtoPool-Concepts + $\mathcal{L}_{\text{Clstk, k=5}}$** | $108 \pm 5$ | $85.5 \pm 0.1$ | $84.4 \pm 0.3$ |
| **ProtoPool-Concepts + $\mathcal{L}_{\text{Clstk, k=10}}$** | $188 \pm 2$ | $85.3 \pm 0.3$ | $85.2 \pm 0.2$ |

# B Training Parameters

In this section we present the exact training hyperparameters, objectives, and schedules used in our experiments for applying the ProtoConcepts module to the ProtoPNet [3], ProtoPool [26], and TesNet [36] prototype networks.

## B.1 ProtoPNet

In addition to the original losses ProtoPNet has, we trained ProtoPNet-Concepts with $\mathcal{L}_{\text{Clstk, k=10}}$ and $\mathcal{L}_{\text{Rad}}$. We set the corresponding weight for losses as demonstrated in Table 6. Like ProtoPNet, we have 5 epochs of training as warmup where we trained the add-on layers, prototype layers, and the radius width, as shown in Table 7. Similar to the original ProtoPNet training schedule, We start training the convolution layer and last layer at the joint optimization stage for 10 epochs. Overall, 15 epochs are trained before pruning and fine-tuning. Lastly, after pruning the prototypical balls that have no visualizations, we fine-tune the last layer for 20 epochs.

Table 6: Parameter Settings for ProtoPNet-Concepts

| Parameter | Weight |
|---|---|
| Cross Entropy Weights | 1.0 |
| Top-$k$ Cluster Loss | 0.8 |
| Separation Loss | $-0.08$ |
| Radius Loss | 0.01 |

Table 7: Training Schedule for ProtoPNet-Concepts

| Training Stage | Model Layers | Learning Rate | Duration |
|---|---|---|---|
| Warmup Stage | add on layers | $3 \times 10^{-3}$ | 5 epochs |
| | prototype vectors | $3 \times 10^{-3}$ | |
| | radius | $0.5 \times 10^{-4}$ | |
| Joint Stage | convolution $f$ | $1 \times 10^{-4}$ | 10 epochs |
| | add on layers | $3 \times 10^{-3}$ | |
| | prototype vectors | $3 \times 10^{-3}$ | |
| | last layer | $1 \times 10^{-4}$ | |
| Finetuning Stage | last layer | $1 \times 10^{-4}$ | 20 epochs |

## B.2 ProtoPool

We trained ProtoPool-Concepts, in addition to its original loss terms, with $\mathcal{L}_{\text{Clstk, k=10}}$ and $\mathcal{L}_{\text{Rad}}$. The corresponding weights for the losses are shown in Table 8. The training schedule is similar to that of the original ProtoPool,

**Table 8: Parameter Settings for ProtoPool-Concepts**

| Parameter | Weight |
|---|---|
| Cross Entropy Weights | 1.0 |
| Top-$k$ Cluster Loss | 0.8 |
| Separation Loss | $-0.08$ |
| Radius Loss | $3 \times 10^{-3}$ |

**Table 9: Training Schedule for ProtoPool-Concepts**

| Training Stage | Model Layers | Learning Rate | Duration |
|---|---|---|---|
| Warmup Stage | add on layers | $1.5 \times 10^{-3}$ | 10 epochs |
|  | pool layer | $1.5 \times 10^{-3}$ |  |
|  | radius | $0.5 \times 10^{-4}$ |  |
| Joint Stage | convolution $f$ | $5 \times 10^{-5}$ | 20 epochs |
|  | add on layers | $1.5 \times 10^{-3}$ |  |
|  | pool Layer | $1.5 \times 10^{-3}$ |  |
| Finetuning Stage | last layer | $1 \times 10^{-4}$ | 15 epochs |

shown in Table 9. We trained radius width only during the warm-up stage. We set the first 10 epochs as the warm-up stage, and the following 20 epochs as the joint optimization stage. Overall, the model is trained for 30 epochs before pruning and fine-tuning. We further trained additional 15 epochs to fine-tune the last layer weights after pruning the prototypical balls that do not contain any visualizations.

## B.3 TesNet

We trained ProtoPool-Concepts, in addition to its original loss terms, with $\mathcal{L}_{\text{Clstk, k=3}}$ and $\mathcal{L}_{\text{Rad}}$. The corresponding weights for the losses are shown in Table 10. The training schedule is similar to that of the original TesNet, shown in Table 11. We trained radius width during the warm-up stage and joint optimization stage. We set the first 5 epochs as the warm-up stage, and the following 15 epochs as the joint optimization stage. Overall the model is trained for 20 epochs before pruning and fine-tuning. We further trained additional 15 epochs to fine-tune the last layer weights after pruning the prototypical balls that do not contain any visualizations.

**Table 10: Parameters Setting for TesNet-Concepts**

| Parameter | Weight |
|---|---|
| Cross Entropy Weights | 1.0 |
| Top-$k$ Cluster Loss | 0.8 |
| Separation Loss | $-0.2$ |
| Subspace Separation Loss | $3 \times 10^{-5}$ |
| Orthogonality Loss | $5 \times 10^{-3}$ |
| Radius Loss | $3 \times 10^{-5}$ |

**Table 11: Training Schedule for TesNet-Concepts**

| Training Stage | Model Layers | Learning Rate | Duration |
|---|---|---|---|
| Warmup Stage | add on layers | $3 \times 10^{-3}$ | 5 epochs |
|  | Prototype Vectors layer | $3 \times 10^{-3}$ |  |
|  | radius | $1 \times 10^{-4}$ |  |
| Joint Stage | convolution $f$ | $1 \times 10^{-4}$ | 15 epochs |
|  | add on layers | $3 \times 10^{-3}$ |  |
|  | Prototype Vectors Layer | $3 \times 10^{-3}$ |  |
|  | radius | $1 \times 10^{-5}$ |  |
| Finetuning Stage | last layer | $1 \times 10^{-4}$ | 15 epochs |

## C  More Examples for ProtoPNet-Concepts Visualizations

In this section, we further demonstrate how multi-visualization solves the ambiguity of prototypes learned by ProtoPNet. The visualizations shown are trained with a VGG-19 [29] backbone for both ProtoPNet and

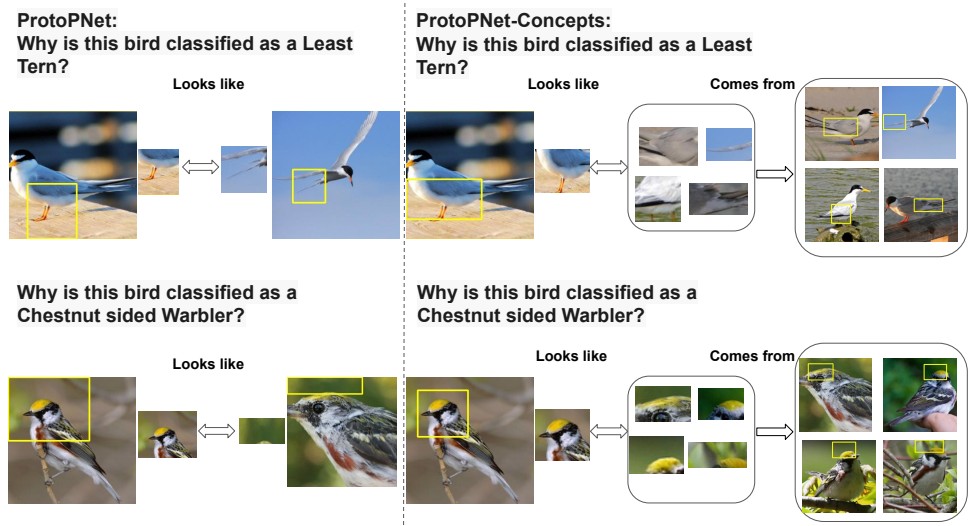

Figure 6: Reasoning of a correctly classified Least Tern (top) and Chestnut-sided Warbler (bottom) by ProtoPNet (left) and ProtoPNet-Concepts (right). Some visualization examples of the Prototypical concepts are shown in the yellow bounding boxes, and compared with patches from the test image.

ProtoPNet-Concepts. Figure 6 demonstrates how our method reduces the ambiguity inherent in the ProtoPNet architecture. The top row provides an example of how ProtoPNet and ProtoPNet-Concepts compare and classify a test image of a Least Tern. ProtoPNet highlights the bird's leg on the test image, which one can barely see in the model's learned prototypical features (e.g., is it comparing the wings? Or is it comparing the feet, which are too tiny to see in the image?) On the other hand, it is clear that ProtoPNet-Concepts is comparing the wings, even though the bounding boxes on the test image include the bird's feet. Similarly, the bottom row shows each model's reasoning for a correctly classified test image of a Chestnut-sided Warbler. It is not clear in this case whether ProtoPNet is comparing the background or a barely-captured yellow crown feature. On the other hand, the many examples of the yellow crown from the multi-visualization of ProtoPNet-Concepts clarify that the prototype represents this crucial feature of a Chestnut-sided Warbler [28].

More examples of unclear comparisons can be found in Figure 7. As shown in the plot, the ProtoPNet finds the green wing and spotted breast of the test image of a spotted catbird similar to a green wing with a black spot in the middle. On the other hand, the prototype learned by ProtoPNet-Concepts suggests that the prototypical concept focuses on the texture. In Figure 7, it is also unclear if ProtoPNet finds the feathers or the red feet of the Forster Tern to be similar to its learned prototype visualizations. However, we can clearly see from the ProtoPNet-Concepts multi-visualizations that the prototypical concept is actually the grey wing and white belly instead of the red feet.

Some types of ambiguities are specific to bird types. For instance, it is difficult to determine what the prototypes represent for birds that are black everywhere. For example, as shown in Figure 8, it is hard to see what the prototypes learned by ProtoPNet are capturing except for the dark feathers. Multiple visualizations learned by ProtoPNet-Concepts suggest that color and possibly the shape of the bird's belly are the first concept, whereas the red eye and shape of the head appear to be the second concept.

Ambiguity of the learned prototypes can arise from low quality test images. For example, as shown in Figure 9, visualizations from ProtoPNet have low brightness/contrast on the prototype birds. It is hard to see any distinguishable features captured by those prototypes. On the other hand, although similar visualizations are used by ProtoPNet-Concepts, other visualizations within the ball demonstrate the prototypical concepts (here, blue and white stripes apparently).

Lastly, having multiple visualizations can also help to draw inferences on misclassified cases. Figure 10 and Figure 11 show cases when both ProtoPNet and ProtoPNet-Concepts misclassify the test image. In Figure 10, it is unclear how ProtoPNet found the test image of a Cedar Waxwing to be similar to a Cardinal, except for the color of the fruit it has in its beak. On the other hand, multiple visualizations offer additional information such as the trianglular silhouette of the head, dark strip of feathers around the eye, and pointed features at the back of the head. Similarly, in Figure 11, ProtoPNet-Concepts is able to offer more specific information than ProtoPNet, such as the grey and white color contrast, black head and the shape of the body. Here, the algorithms are showing us why it is difficult to tell these birds apart.

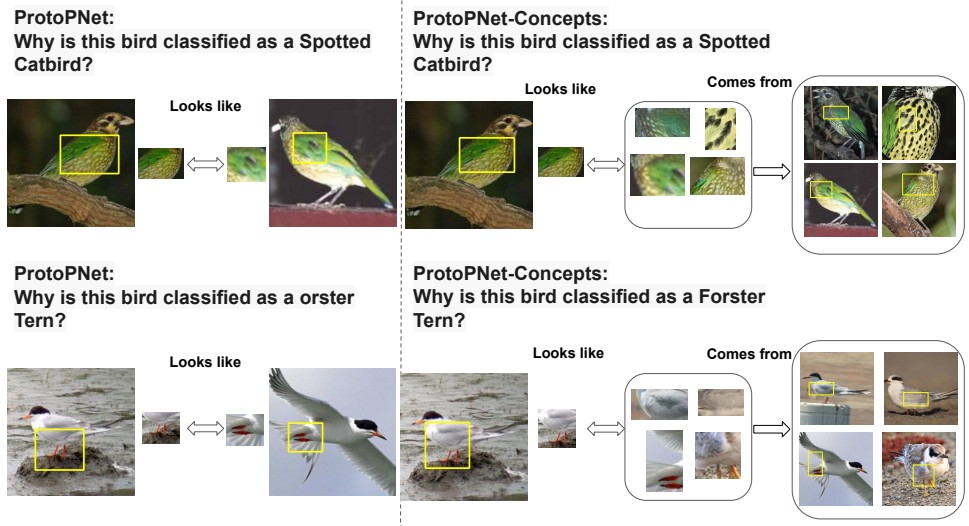

Figure 7: Reasoning of a correctly classified Spotted Catbird (top) and Forster Tern (bottom) by ProtoPNet (left) and ProtoPNet-Concepts (right).

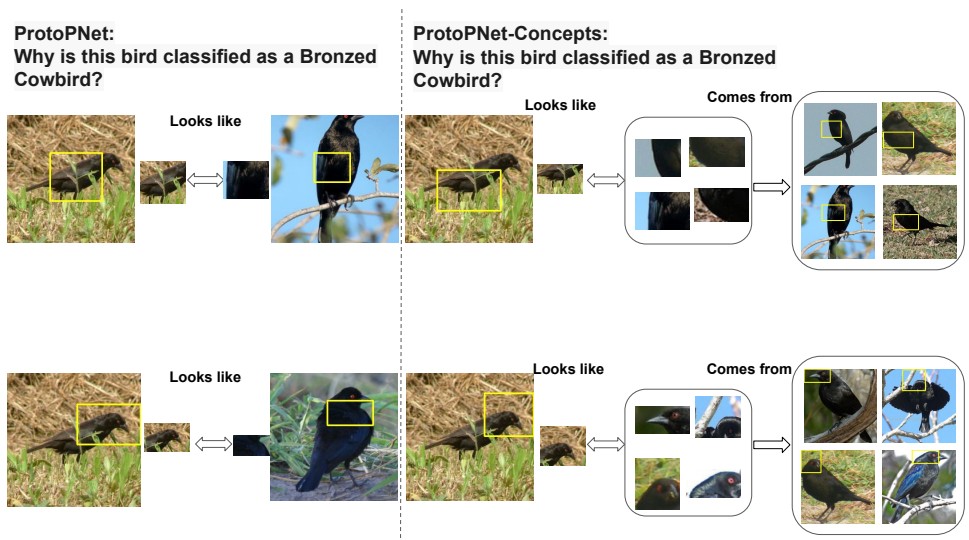

Figure 8: Reasoning of a correctly classified Bronzed Cowbird by ProtoPNet (left) and ProtoPNet-Concepts (right).

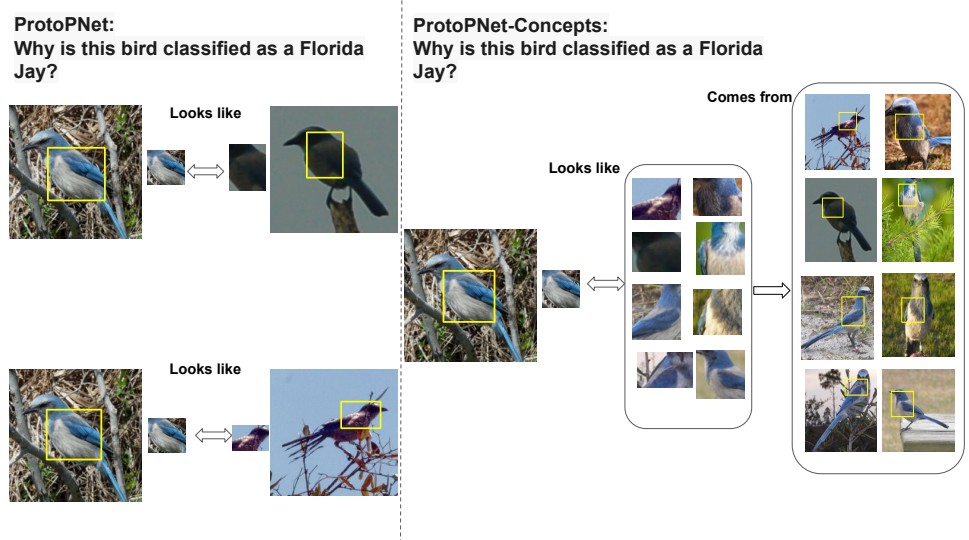

Figure 9: Reasoning of a correctly classified Florida Jay by ProtoPNet (left) and ProtoPNet-Concepts (right).

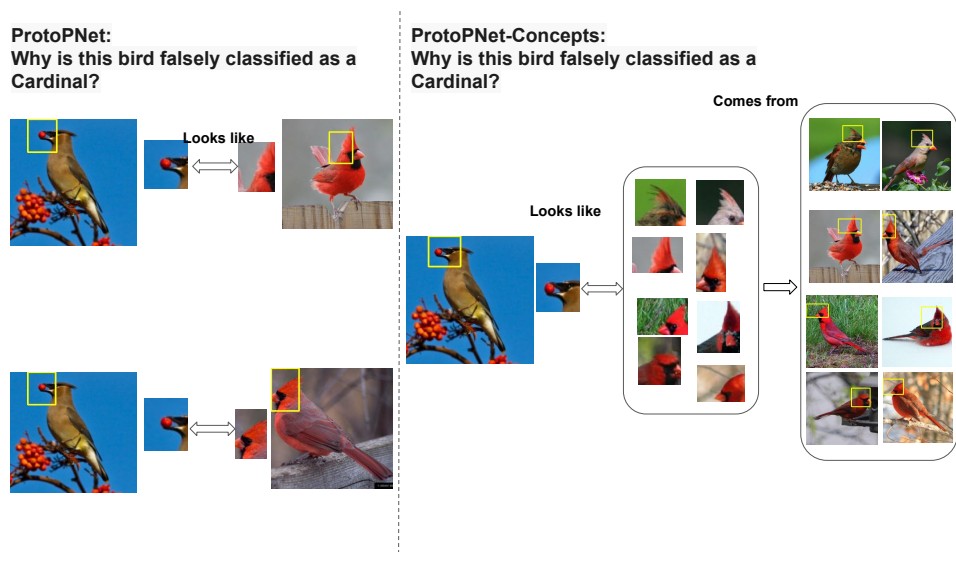

Figure 10: Reasoning of a Misclassified Cedar Waxwing to Cardinal by ProtoPNet (left) and ProtoPNet-Concepts (right).

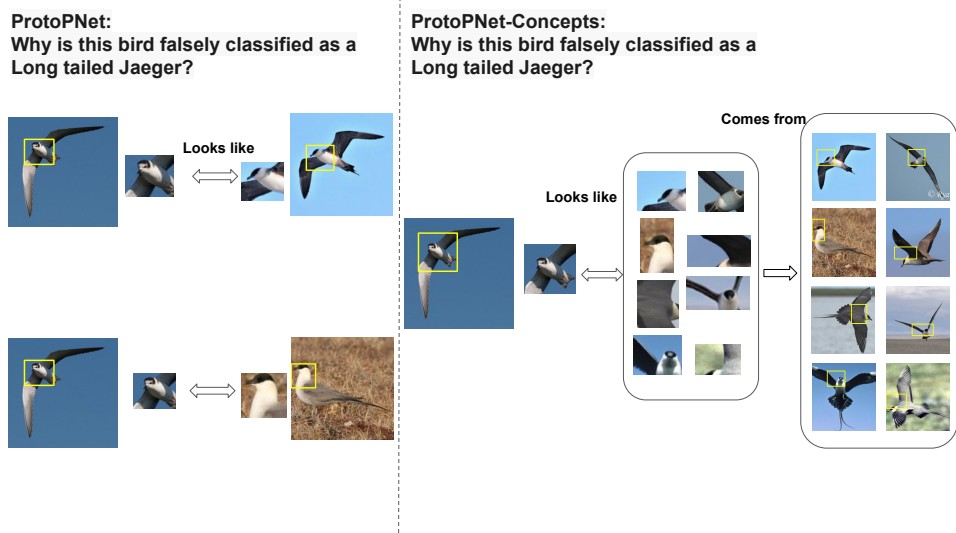

Figure 11: Reasoning of a Misclassified Black Tern to Long Tailed Jaeger by ProtoPNet (left) and ProtoPNet-Concepts (right).

# D More Examples for ProtoPool-Concepts Visualizations

In this section, we present more examples of the reasoning process for ProtoPool-Concepts. The visualizations are obtained from the model with ResNet 50(iNat) [10] backbone. Because prototypes are shared among different classes, the information extracted by ProtoPool can be unintuitive, especially when the prototype does not come from the same class as a given test image. This problem affects ProtoPool as well as ProtoPool-Concepts – while shared prototypes can, in some cases, make a concept clearer by borrowing good examples of that concept from other classes, in our experience it seems to make the concepts less clear. For example, in Figure 12, it is hard to see how the beak of a Cedar Waxwing is similar to the prototype visualization captured by ProtoPool. Moreover, similar to a problem we saw with ProtoPNet, whether the prototype is capturing the feet or the belly of the Cedar Waxwing from the top row is also unclear. On the other hand, ProtoPool-Concepts may be capturing a color contrast of the throat and breast that is common also to the Great Crested Flycatcher and Cedar Waxwing, etc., but since the contrast of the test image is not particularly good, the human might determine not to trust that

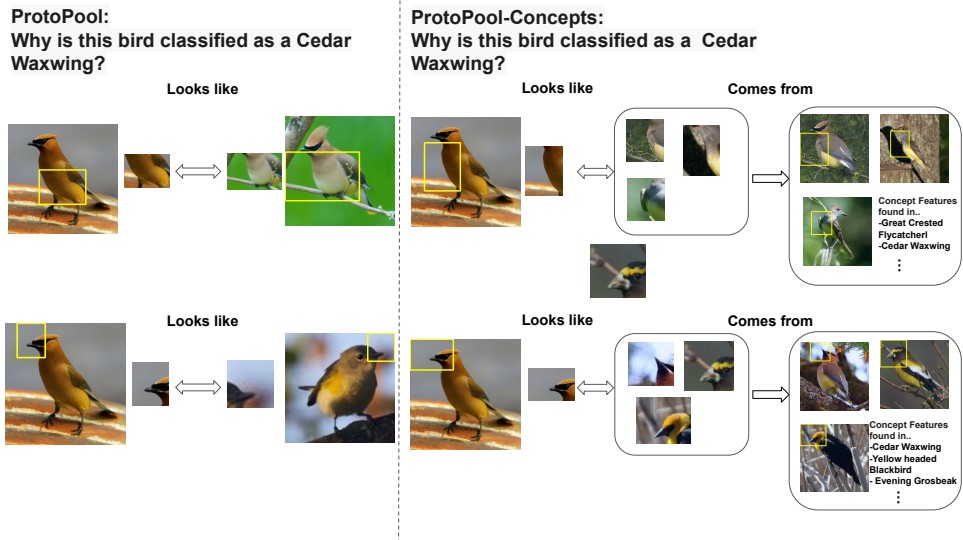

Figure 12: Reasoning of a correctly classified Cedar Waxwing by ProtoPool (left) and ProtoPool-Concepts (right).

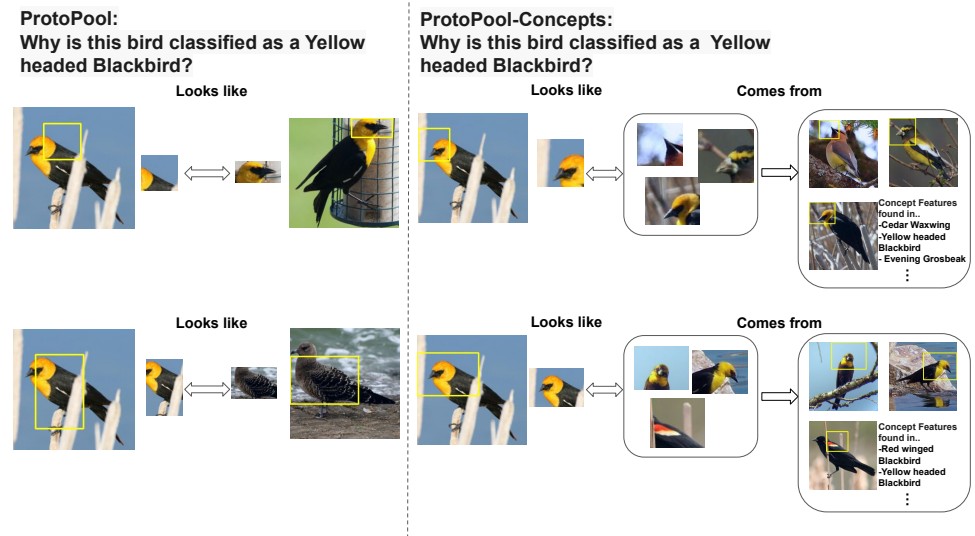

Figure 13: Reasoning of a correctly classified yellow-headed blackbird by ProtoPool (left) and ProtoPool-Concepts (right).

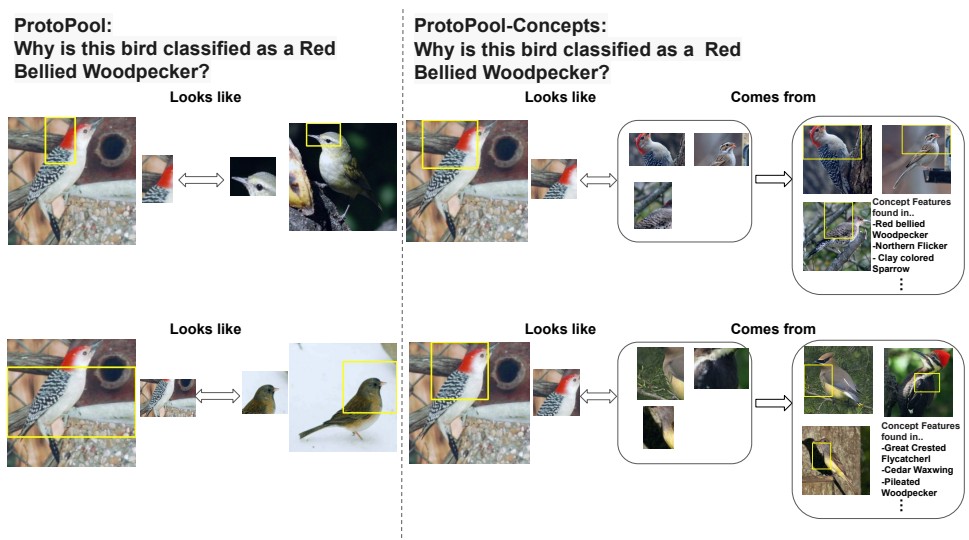

Figure 14: Reasoning of a correctly classified Red Bellied Woodpecker by ProtoPool (left) and ProtoPool-Concepts (right).

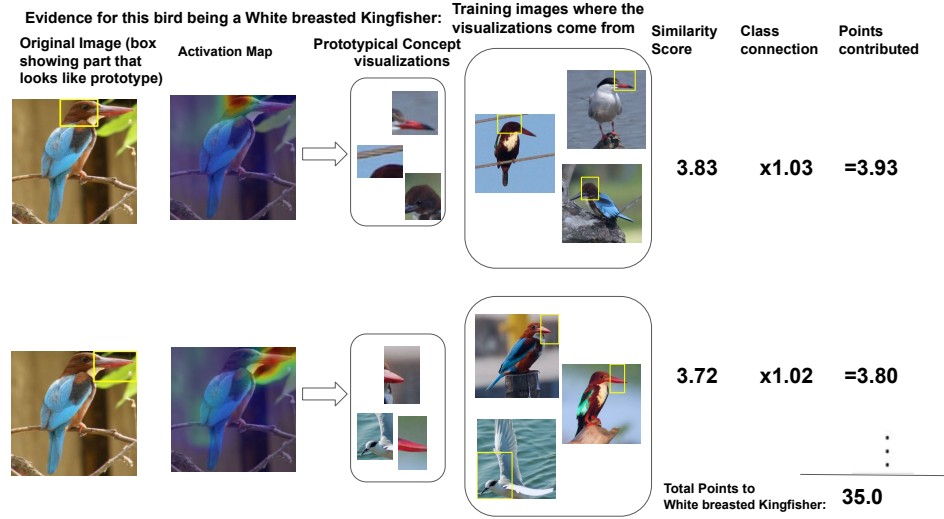

Figure 15: Reasoning of a correctly classified White-breasted Kingfisher by ProtoPool-Concepts

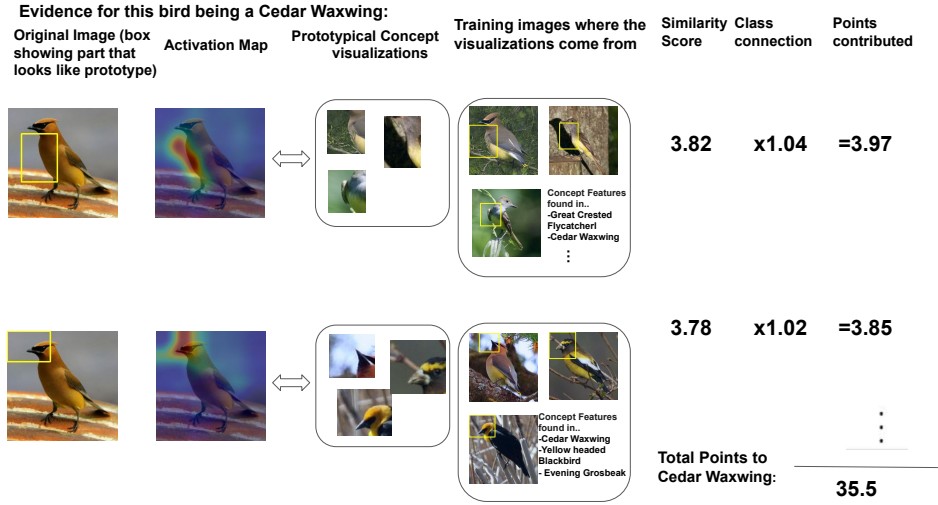

Figure 16: Reasoning of a correctly classified Cedar Waxwing by ProtoPool-Concepts

comparison. ProtoPool-Concepts also appears to capture the yellow and black color pattern on the bird's head. Similarly, in Figure 13, the comparison made at the bottom left for ProtoPool is hard to identify with a single visualization, while ProtoPool-Concepts offers more information such as the yellow head and black body, though the concept is not cohesive, since it includes a bird with red and white stripes. An analyst seeing this concept might determine that this concept may not be trustworthy, which is a useful piece of information. It is worth noting that the prototypical concept of the yellow-headed blackbird shown at the top of Figure 13 is shared with the Cedar Waxwing as shown in the bottom row of Figure 12. Figure 14 shows that ProtoPool-Concepts shares part of a prototype concept with the Cedar Waxwing; this concept appears to indicate a dark-and-light textural pattern that is similar to the wing of the Red Bellied Woodpecker. However, the comparison made by ProtoPool is ambiguous. Figure 15, Figure 16 further demonstrate how the evidence layer and prototype activation are involved in the reasoning process. As explained in the main paper, a weighted score would be computed for each of the prototypical concepts, and a total logit score would be calculated for the final prediction.

# E    More Examples for TesNet Visualizations for CUB dataset

In this section, we present some examples of the reasoning process of TesNet-Concepts and its comparison with TesNet. The visualizations were obtained from TesNet and TesNet-Concepts trained with a VGG19 backbone. Similar to ProtoPNet and ProtoPool, ambiguity arises from low quality images. In Figure 17, the prototype visualization of TesNet for a Florida Jay shown at the top has low contrast. Whether the prototype is capturing the belly or the feet is unclear. Although a similar prototype visualization is also captured by the prototypical ball for TesNet-Concepts as shown in the top right of the plot, the other visualizations reveal the prototypical concept as the white belly near the blue feathers. Similarly, the multi-visualization shown at the bottom suggests that the prototypical ball likely represents the bird's head and beak. Figure 18 shows the prototype logic for identifying a Cedar Waxwing by the color gradient on the bird's throat. Some of the image patches chosen by TesNet-Concepts look almost identical to the chosen patch of the test image; this is more convincing than the comparisons from TesNet without concepts.

# F    More Examples for ProtoPool-Concepts Visualizations and Model Performance for Car dataset

In this section, we present some of the visualizations for ProtoPool-Concepts on the Car dataset. The visualizations are from the models trained with a DenseNet121 [11] backbone. Similar to ProtoPool-Concepts for the CUB 200-2011 dataset, the shareable prototypes still make the comparison unintuitive. As shown in Figure 19, ProtoPool seems to compare the front of a BMW to that of an Audi. It appears to find a similarity between the kidney grille of the BMW to the front of the Audi, or it could be the shape of the lights that are similar. On the other hand, the multiple visualizations from ProtoPool-Concepts suggest that the model is actually capturing the kidney grille of the BMW. The other visualizations further show that the ProtoPool-Concept found a part of the car on the bottom, just in front of the tires, is similar to the same part of the other types of SUVs. Figure 20 shows that our model finds the wheel of the Audi TT RS Coupe is similar to the wheels of different types of Audis.

Table 12 shows some additional performance experiments for ProtoPNet-Concepts and ProtoPool-Concepts on the Cars dataset. The training schedule and parameter settings are the same as reported above. The initialized number of prototypes for ProtoPNet-Concepts and ProtoPool-Concepts are 1960 and 195 respectively. The radius initialization for ProtoPNet-Concept is 7.5, and the radius initialization for ProtoPool-Concept is 4.7. As shown in the table, ProtoPNet-Concepts under the DenseNet121 baseline has better performance with fewer visualizable prototypes than ProtoPNet, and its performance is very close to the baseline. On the other hand, ProtoPool-Concepts under a DenseNet161 backbone yields better performance than ProtoPNet with much fewer visualizable prototypes.

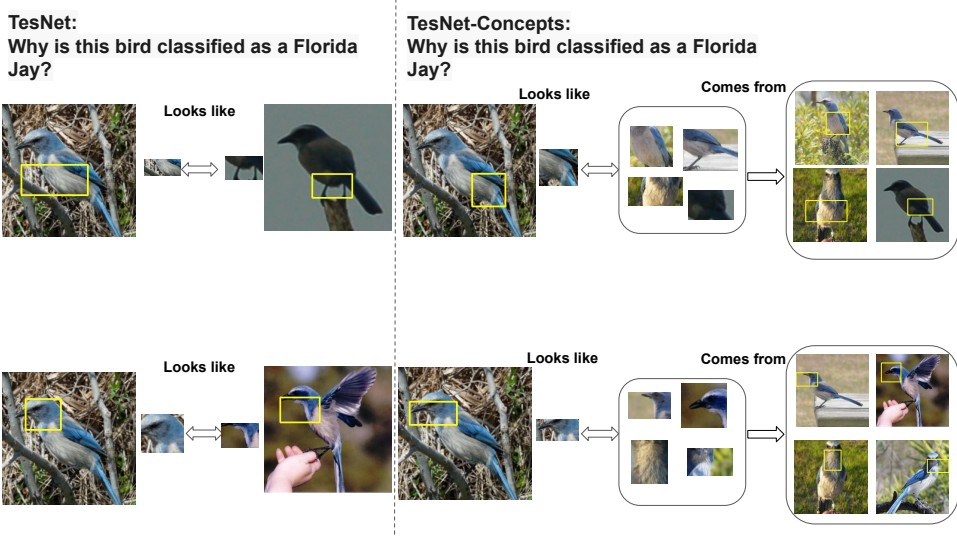

Figure 17:   Reasoning of a correctly classified Florida Jay by TesNet (left) and TesNet-Concepts (right)

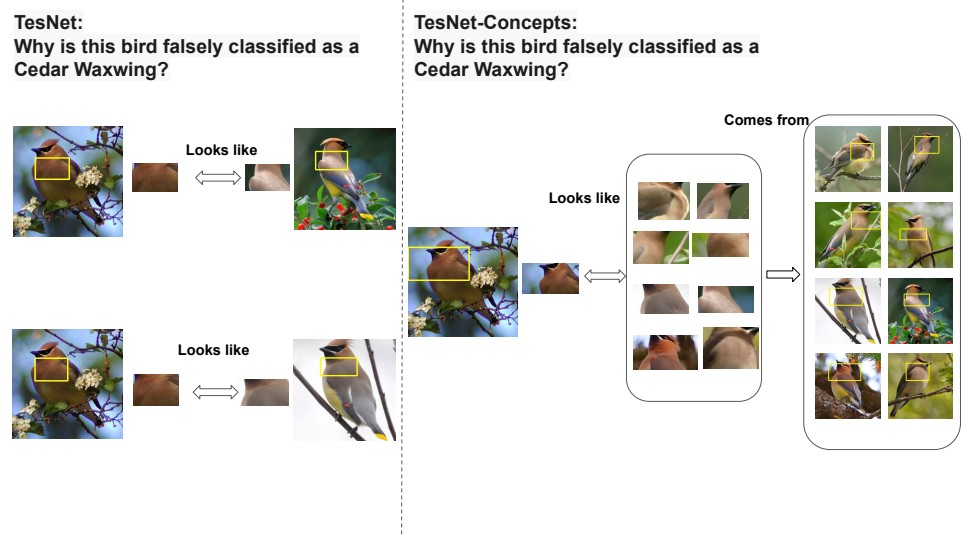

Figure 18: Reasoning of a correctly classified Cedar Waxwing by TesNet (left) and TesNet-Concepts (right)

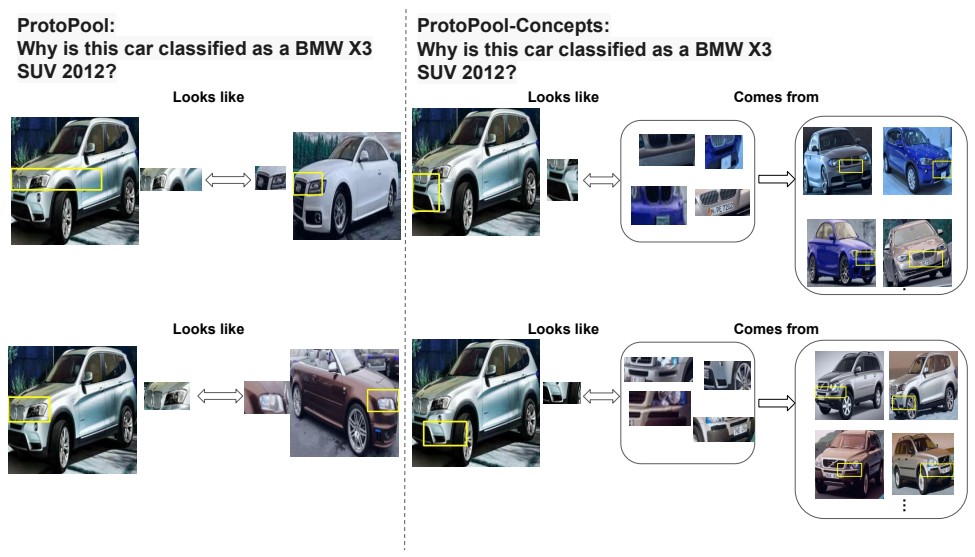

Figure 19: Reasoning of a correctly classified BMW X3 SUV 2012 by ProtoPool (left) and ProtoPool-Concepts (right).

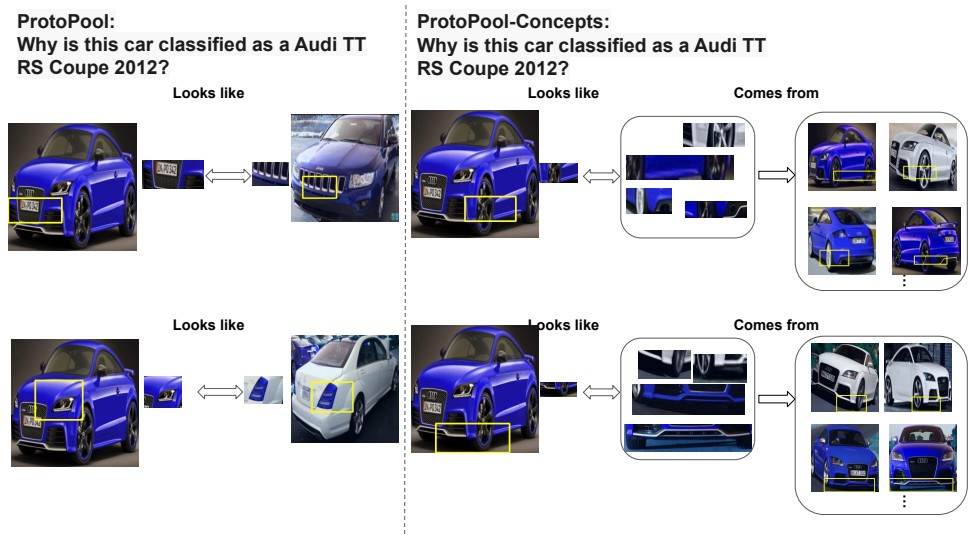

Figure 20: Reasoning of a correctly classified Audi TT RS Coupe 2012 by ProtoPool (left) and ProtoPool-Concepts (right).

Table 12: Accuracy comparison of ProtoPool-Concepts to other shared prototype models on the Stanford Cars dataset

| Arch. | Model | Prototype $p$ # | Acc.[%] |
|---|---|---|---|
| | **ProtoPNet-Concepts (ours)** | $1941 \pm 7$ | $88.4 \pm 0.1$ |
| DenseNet121 [11] | ProtoPNet [3] | 1960 | $86.8 \pm 0.1$ |
| | Baseline (given in [3]) | N/A | $89.7 \pm 0.1$ |
| | **ProtoPool-Concepts (ours)** | 195 | $90.4 \pm 0.4$ |
| DenseNet161 [11] | ProtoPNet [3] | 1960 | $89.5 \pm 0.2$ |
| | Baseline (given in [36]) | N/A | $92.5 \pm 0.3$ |

## G Training software and platform

We implemented our ProtoConcepts on ProtoPNet, ProtoPNet, and ProtoPool using PyTorch. The experiments were run on 1 NVIDIA Tesla V100 or 1 NVIDIA RTX A5000. Using the ProtoConcept module with any base architecture has similar training times to using the base architecture alone.

