# OpenReview forum: "This Looks Like Those: Illuminating Prototypical Concepts Using Multiple Visualizations"
_NeurIPS.cc/2023/Conference — NeurIPS 2023 poster_

### Official Review · Reviewer_x5Kg · 2023-06-21

**Soundness:** 3 good
**Presentation:** 3 good
**Contribution:** 2 fair
**Rating:** 6
**Confidence:** 5

**Summary:**

This paper proposes a new interpretable prototype-based classifier, called ProtoConcepts. Unlike existing prototype-based classifiers that use one-to-one comparisons, ProtoConcepts learns prototypical concepts using multiple image patches. This approach aims to make it easier to identify the underlying concept being compared, allowing for richer and more interpretable visual explanations. Experiments show that this modified "this looks like those" reasoning process can be applied to various prototypical image classification networks without affecting accuracy on benchmark datasets.


**Strengths:**

1.	The authors make a valid point that utilizing a single training image patch as a prototype can be insufficient for users to comprehend the concept the prototype represents. For example, when presented with a "blue circle" prototype, it is unclear whether the concept is related to the color "blue" or the shape "circle."

2.	The paper (Table 1) demonstrates that ProtoConcepts can be integrated with various prototype-based classifiers, such as ProtoPNet, TesNet, and ProtoPool, showcasing its adaptability across different methods.


**Weaknesses:**

1. It is definitely not a new idea in the XAI community to use multiple images/patches to visualize a concept. This is a common practice in the works of concept-level explanations, like [TCAV](http://proceedings.mlr.press/v80/kim18d/kim18d.pdf), [ACE]( https://arxiv.org/pdf/1902.03129.pdf), and so on. However, none of them are discussed in the paper;


2. The main issue of the paper lies in the evaluation of interpretability. It is unclear whether these examples are cherry-picked. Also, given that the source code of the method is unavailable at this moment, it is unknown how good/bad the explanation results would be for many other cases.

If the examples are not cherry-picked, it would be better for the authors to clearly state this in the paper or use examples in a fixed order (e.g., always use the first example of each class).

It is hard to be convinced that the interpretability of ProtoConcepts is good by showing several separate examples. I would suggest that the authors conduct a carefully designed human user study. The user study results should better convince users that the method brings about a better understanding of the model's behavior. User studies are often conducted in the work of concept-level explanations.

**Questions:**

See weaknesses.



**Limitations:**

1. The novelty of the paper appears to primarily lie in the combination of two distinct lines of research within XAI - concept-level explanations and prototype-based classifiers.

2. The absence of user study results and the unavailability of source code make it difficult to be convinced about the interpretability of the explanations generated by the proposed method.

See more details in the part of "Weaknesses".

---

> ### Author Rebuttal · Authors · 2023-08-10
>
> Thank you for your review comments. We are happy to make some clarifications as follows:
> >It is definitely not a new idea in the XAI community to use multiple images/patches to visualize a concept. This is a common practice in the works of concept-level explanations, like TCAV, ACE, and so on. However, none of them are discussed in the paper
>
> **A**:Thank you for bringing these relevant works to our attention; we agree that concept-level explanations are not a new idea in the field of XAI, and will add discussion of these works to our final manuscript. However, TCAV and ACE are from an entirely different genre than our approach. Those methods are **posthoc** and **supervised**. In other words, TCAV needs to be fed images of a **known, predefined** concept and used those to analyze an existing model. Unlike our method, **posthoc** methods are not involved in the model computation process. Any results produced **posthoc** are not always faithful to the classification decision. Instead our method is **inherently interpretable**, and the algorithm **discovers the concepts** by itself. It is similar to ProtoPNet and its variants, but none of those learn concepts; they use comparisons between **pairs** of images. The concepts that ProtoConcepts discovers could be totally new, whereas TCAV/ACE/etc. require **known** concepts.
> >I would suggest that the authors conduct a carefully designed human user study.
>
> **A**:As per your suggestion, we evaluated the added interpretability of our ProtoPConcepts module using the **distinction** survey task in HIVE[1]. In this task, we asked participants to select the correct prediction out of two options based on provided visual explanations, and found that our ProtoPConcepts module allowed for a statistically significant increase in user interpretability. Please see the global rebuttal for exact study details and results. We thank for the suggestion of a human study and believe that our survey results objectively show the interpretability benefit from our method.
>
> >If the examples are not cherry-picked, it would be better for the authors to clearly state this in the paper or use examples in a fixed order (e.g., always use the first example of each class).
>
> **A**:Thank you for this comment. We believe the results are representative. Luckily, you can judge yourself whether you believe the results are cherry picked because we included many additional examples of the reasoning process in the supplement. In addition, our user study shows that our method results in an objective increase in user interpretability of our visual explanations.
>
> [1] Kim et al., HIVE: evaluating the human interpretability of visual explanations, ECCV 2022

---

> > ### Comment · Reviewer_x5Kg · 2023-08-12
> > **Response to Rebuttal**
> >
> > I appreciate the authors' response.
> >
> > I take the point that the concept-level explanation methods like TCAV and ACE are posthoc methods while the proposed method in the paper is inherently interpretable. However, the work of concept-level explanation methods should definitely be discussed in the final version of the paper. Also, note that while TCAV requires known concepts, ACE does not.
> >
> > The human study results supplied are vital in persuading readers about the interpretability of ProtoConcepts. The results should also be included in the final version.
> >
> > I raised my rating of the paper to 6.

---

> > > ### Author Response · Authors · 2023-08-14
> > >
> > > Thank you for the correction on ACE. However, it is still a post-hoc method. It would require an outlier removal step to "make every cluster of segments clean of meaningless or dissimilar segments"[1]. It also relies on segmentation methods “at the cost of suffering from lower segmentation quality”[1]. We believe it’s a valuable discussion, and we will make sure to include it in the final manuscript. Thank you again for pointing it out. And we will definitely add the user study results to the main manuscript. Thank you again.
> > >
> > > [1]Ghorbani, Amirata, et al. "Towards automatic concept-based explanations." Advances in neural information processing systems 32 (2019).

---

### Official Review · Reviewer_wBiV · 2023-07-03

**Soundness:** 2 fair
**Presentation:** 3 good
**Contribution:** 3 good
**Rating:** 6
**Confidence:** 4

**Summary:**

The paper proposes a modification of the prototype layer of the existing prototype-based networks. Existing methods rely on a single training image patch as a prototype. The main drawback of this is that it is difficult for the user to understand the meaning of the prototype from a single visualization. The paper proposes to learn prototypical concepts which are visualized using multiple training image patches. In this way, the semantic of each prototype is less ambiguous.

**Strengths:**

_Clarity_: the paper is generally well-written and structured clearly. The figures showing image examples are very helpful, especially for readers that are not experts on prototypical networks.

_Significance_: the paper addresses a relevant problem which is well known to practitioners using prototypical networks. The problem is filling the gap between what the prototypes represent and the human understanding of prototype semantics. Previous works tried to tackle it.
The proposed changes do not badly impact the overall performance of the prototypical networks; indeed, the accuracy is comparable to previous methods.

_Quality and originality_: the two novel losses are technically sound. The idea of representing the prototypes as a ball in the latent space is interesting. The solution can be applied to several types of prototypical networks.

_Reproducibility_: the code will be made available upon acceptance (it was unavailable to the reviewers). The supplementary material reports the training parameters.

**Weaknesses:**

As reported in the introduction, the proposed solution should allow the user to determine the semantic meaning of each prototype with less ambiguity. However, the experiments do not report one or more metrics that show this reduction of ambiguity with respect to the plain prototypical networks. The paper and the supplementary material reports some example of concrete cases but not an overall evaluation. The optimal solution would be to run an evaluation with real users, but I understand that it is expensive.
The paper [1] proposes activation precision as an interpretability metric (section 4.2) (the segmentation masks are provided with CUB-200).

Given that Cub200 dataset provides part location annotations for the images, another metric could measure the degree to which the multiple visualizations of the prototype refer to the same bird part.

[1] Barnett 2021, "IAIA-BL: A Case-based Interpretable Deep Learning Model for Classification of Mass Lesions in Digital Mammography"

**Questions:**

- what is the main advantage of your solution with respect to learning standard prototypes (e.g., as done in ProtoPNets) and visualizing it on multiple most activated training images? In other words, the original work on ProtoPNets visualizes the prototype on the most activated training image; what if you just take more images?
- How is k set in the experiments? Why does the value of k change depending on the prototype-based model?

**Limitations:**

The authors address the limitations. I would like to stress a point that is also discussed in the limitations. The user expertise influences the impact of the method on identifying the concepts: given the same set of prototype visualization, an expert user has a different interpretation of them with respect to a non-expert.

---

> ### Author Rebuttal · Authors · 2023-08-10
>
> Thank you for your review comments. We are happy to make some clarifications as follows:
>
> >However, the experiments do not report one or more metrics that show this reduction of ambiguity with respect to the plain prototypical networks.
>
> **A**: As per your suggestion, we evaluated the added interpretability of our ProtoPConcepts module using the **distinction** survey task in HIVE[1]. In this task, we asked participants to select the correct prediction out of two options based on provided visual explanations, and found that our ProtoPConcepts module allowed for a statistically significant increase in user interpretability. Please see the global rebuttal for exact study details and results. We thank for the suggestion of a human study and believe that our survey results objectively show the interpretability benefit from our method.
>
> >What is the main advantage of your solution with respect to learning standard prototypes (e.g., as done in ProtoPNets) and visualizing it on multiple most activated training images? In other words, the original work on ProtoPNets visualizes the prototype on the most activated training image; what if you just take more images?
>
> **A**:Unfortunately, the type of calculation you're thinking about doesn't work for models such as ProtoPNet, and is part of the motivation for this work. The reason why models such as ProtoPNet are interpretable-by-design is that the prototypes learned by the model correspond **directly** to a single training patch in latent space. When models such as ProtoPNet perform inference on a test image, they calculate distances to these exact training patches in latent space. Therefore, the patches visualized by ProtoPNet are not just the **closest** training patch to that prototype, they are **exactly** the single training patch corresponding to that prototypical vector in latent space. If we tried to create a way to highlight other nearby training patches, it would yield unfaithful explanations, as the distances calculated in the model's reasoning process doesn't directly correspond to other nearby parts. To overcome this problem, we instead represent prototypes as sets for both the reasoning and explanation process, which allows us to create multiple visualizations for each prototype in a way that is still faithful to the model's exact reasoning process.
> >How is k set in the experiments? Why does the value of k change depending on the prototype-based model?
>
> **A**:In the top-k cluster loss for our optimization, the exact value of k is treated as a hyperparameter and is selected by cross-validation. Because the prototype-based models we analyze in our experiments have widely varying latent space geometries due to different distance metrics (e.g. L2 distance vs. cosine similarity), we tune the exact value of k separately for these different models along with other parameters such as the initial concept ball radius.
>
> [1] Kim et al., HIVE: evaluating the human interpretability of visual explanations, ECCV 2022

---

> > ### Comment · Reviewer_wBiV · 2023-08-14
> > **Response to rebuttal**
> >
> > Thank you for your answer and for taking the time to run a user study.
> >
> > The user study improves the evaluation part. I agree with Reviewer x5Kg that the user study should be added to the final version of the main text.
> >
> > I agree with your explanation about visualizing the prototypes on multiple most activated training images. Would it make sense to have an additional experiment to demonstrate this also visually by showing the user the same number of visual explanations for both ProtoPNets and ProtoConcept? I expect that ProtoConcept wins the comparison because its multiple visualizations are faithful and of higher quality. It is not only a matter of having more multiple visualizations but also of their quality and interpretability. This can also be shown by computing the activation precision metric (see IAIA-BL paper), which should be higher for ProtoConcept. This may partially address the charry-picking issue raised by Reviewer p3JV and Reviewer x5Kg.
> >
> > I raised the score to 6.

---

> > > ### Author Response · Authors · 2023-08-14
> > >
> > > Thanks for your response! We will definitely add the user study results to the final manuscript, as you and other reviewers have suggested.
> > >
> > > >Would it make sense to have an additional experiment to demonstrate this also visually by showing the user the same number of visual explanations for both ProtoPNets and ProtoConcept?
> > >
> > > **A**: In regards to comparing multiple visualizations between ProtoPNet and ProtoConcepts, we can add a comparison of the multiple visualizations of ProtoConcepts to the **closest patch** visualizations of ProtoPNet to the supplement, noting that the **multiple visualizations** generated for ProtoPNet are not the actual patches used in inference, but rather generated as a visual comparison of quality and interpretability as you suggest, similar to in Fig. 5 of ProtoPNet paper[1].
> > >
> > > > This can also be shown by computing the activation precision metric (see IAIA-BL paper), which should be higher for ProtoConcept.
> > >
> > > **A**:Unfortunately, since the activation precision metric requires bounding box part annotations and all of the prototypes learned by ProtoPNet and ProtoConcepts for the CUB dataset are learned from the data without explicit part supervision, there isn't a straightforward way for us to compute the activation precision metric for either the CUB or Stanford Cars datasets used in our experiments.
> > >
> > > Although CUB contains part annotations, we do not use them during training, so neither ProtoPNet nor ProtoConcepts are trained to learn the parts corresponding to these annotations explicitly. Specifically, our visualizations are not always restricted to the specific part of the bird, e.g., only the belly or only the neck. It could include the part of the neck and bellow, as shown in Supplementary Figure 1. Thus, it is hard to compute the activation precision metric based on that.
> > >
> > > IAIA-BL was trained using expert annotations on exactly where the network is allowed to activate[2]. Such annotations don’t really make sense for the bird dataset since the concepts could be (as mentioned) at the boundary between the bellow and the neck, which doesn’t have a name and wouldn’t be annotated by a human (but is still a good concept for us to have discovered and used).
> > >
> > > [1] Chaofan Chen, Oscar Li, Daniel Tao, Alina Barnett, Cynthia Rudin, and Jonathan K Su. This looks like that: deep learning for interpretable image recognition. Advances in Neural Information Processing Systems, 32, 2019
> > >
> > > [2] Alina Jade Barnett, Fides Regina Schwartz, Chaofan Tao, Chaofan Chen, Yinhao Ren, Joseph Y. Lo, and
> > > Cynthia Rudin. IAIA-BL: A case-based interpretable deep learning model for classification of mass lesions
> > > in digital mammography. Nature Machine Intelligence, 3:1061–1070, 2021.

---

### Official Review · Reviewer_p3JV · 2023-07-06

**Soundness:** 3 good
**Presentation:** 2 fair
**Contribution:** 2 fair
**Rating:** 6
**Confidence:** 4

**Summary:**

This paper extends the patch comparison of prototypical part-based classifiers from one-to-one patch to one-to-multiple (e.g. from this looks like that to this looks like those). They compare their proposed network with Proto-PNet variants and show no classification accuracy improvements and little evidence of better interpretability. The proposed method was tested on CUB and Cars as previously done in ProtoPNet variants.

I have read the author’s rebuttal and adjust the rating accordingly!

**Strengths:**

1. The research is well positioned in the literature of prototypical part-based classifiers.
2. The extension from 1vs1 to 1vsmany is interesting and could help improve human understanding tasks that all prototypical part-based classifiers fall short on [1]

[1] hive: evaluating the human interpretability of visual explanations

**Weaknesses:**

There are multiple weaknesses of this paper, mostly about the novelty.
1. I see the extension is incremental though well motivated. There is no surprise in the classifier accuracy compared to other ProtoPNet variants. I always see (all of them) they perform roughly the same. I believe the "big fish" is to really make this prototypical part-based explanations usable for humans.
2. Although it was motivated by the lack of interpretability of prototypical part-based classifiers, the analysis of model interpretability is superficial and not spot-on. I believe maybe this work also fails to help humans [1] as its siblings [2,3]. The authors could improve the interpretability analysis by running more extensive study (both automatic and human studies) rather than dissecting sample-wise visualizations that could be easily cherry-picked and misled.
3. I think the paper lacks the analysis of computation cost (i.e. 1vs1 and 1vsmany comparison).

[2] ProtoPNet
[3] ProtoTree

**Questions:**

The 1vsmany patch-wise comparison has been studies and they even conducted human studies on interpretability. How this work differs from them [4].

[4] visual correspondence-based explanations improve ai robustness and human-ai team accuracy

---

> ### Author Rebuttal · Authors · 2023-08-10
>
> Thank you for the review. However, we would like to make some clarifications as following:
> >I believe maybe this work also fails to help humans [1] as its siblings [2,3]. The authors could improve the interpretability analysis by running more extensive study (both automatic and human studies) rather than dissecting sample-wise visualizations that could be easily cherry-picked and misled.
>
> **A**: As per your suggestion, we evaluated the added interpretability of our ProtoPConcepts module using the **distinction** survey task in HIVE[1]. In this task, we asked participants to select the correct prediction out of two options based on provided visual explanations, and found that our ProtoPConcepts module allowed for a statistically significant increase in user interpretability. Please see the global rebuttal for exact study details and results. We thank for the suggestion of a human study and believe that our survey results objectively show the interpretability benefit from our method.
>
> In terms of cherry-picking, we included a lot of examples in the appendix so the user can see whether we are cherry picking (we are not).
>
> >I think the paper lacks the analysis of computation cost
>
> **A**: Thank you for this input; we find that our method is actually more computationally efficient than previous prototype-based methods. Although our method allows for multiple visualizations, the only real computational overhead that is added comes when generating visual explanations after inference by calculating which training patches lie within each concept ball. However, this operation only needs to be performed once and is still relatively fast. During training and inference, the addition of our ProtoConcepts module amounts to a single thresholding layer after the distance calculation in latent space, which we have found to have negligible computational cost. In addition, our set representation for prototypical concepts allows us to skip the projection step of other prototype-based networks, which allows for faster training of a ProtoPConcepts network. We thank the reviewer again for this suggestion and will add further discussion of computational efficiency to the supplement.
>
> >The 1vsmany patch-wise comparison has been studies and they even conducted human studies on interpretability. How this work differs from them [4].
>
> **A**:There are some major differences between our work and [4]. [4] do not learn any concepts or prototypes. Instead, their study uses kNN-based modeling and has to compare a given test image with all training images to find the top 20 training images that are most similar to the test image. This is computationally expensive. On the other hand, our work does not use kNN-based modeling and instead learns a fixed set of relevant prototypes and concepts from the training set. Unlike [4], our model works by comparing a given test image with the learned prototypes/concepts, thereby avoiding having to compare each test image with the entire training set. We also benefit from learning concepts that are meaningful in the domain. Although the study by [4] also considers one vs. multi-patches algorithm, the multiple visualizations are ranked by their similarity scores to the given test image and thus contribute unevenly to the decision process. However, our approach finds multiple visualizations for each learned concept which contribute **equally** to the decision process.
>
> [1] Kim et al., HIVE: evaluating the human interpretability of visual explanations, ECCV 2022
>
> [2] Chaofan Chen, Oscar Li, Daniel Tao, Alina Barnett, Cynthia Rudin, and Jonathan K Su. This looks like that: deep learning for interpretable image recognition. Advances in Neural Information Processing Systems, 32, 2019
>
> [3] Meike Nauta, Ron Van Bree, and Christin Seifert. Neural prototype trees for interpretable fine-grained image recognition. In Proceedings of the IEEE/CVF Conference on Computer Vision and Pattern Recognition, pages 14933–14943, 2021.
>
> [4] Nguyen, Giang, Mohammad Reza Taesiri, and Anh Nguyen. "Visual correspondence-based explanations improve AI robustness and human-AI team accuracy." Neural Information Processing Systems (NeurIPS) (2022).

---

> > ### Comment · Reviewer_p3JV · 2023-08-13
> > **Concerns remained about the evaluation and novelty**
> >
> > I would like to thank the authors for the efforts in the rebuttal. After reading your rebuttal and reviews from other reviewers, my replies are:
> >
> > 1. Not only me, the other reviewers also question on the novelty of this work (wBiV in Q1, x5Kg in W1).
> > 2. I genuinely value the authors' efforts in conducting the human study, but I remain unconvinced by the results.
> > * The report lacks specifics regarding the human study—like the number of participants and their expertise (whether they were experts or laypeople), etc…
> > * Your comparison with the random choice (50%) and ProtoPNet—a model with nearly random choice accuracy—doesn’t adequately capture human utility due to the weak baselines employed.
> >
> > Given that, my primary concerns are still about the novelty and evaluation methodology of this paper.
> > I raise the score (to 4) and am open for more discussion.

---

> > > ### Author Response · Authors · 2023-08-14
> > >
> > > Thank you for your comments, and no problem, let's try again.
> > > >Not only me, the other reviewers also question on the novelty of this work (wBiV in Q1, x5Kg in W1).I genuinely value the authors' efforts in conducting the human study, but I remain unconvinced by the results.
> > >
> > > **A**: Our paper's novelty is to provide a **non-posthoc** method that discovers interpretable concepts and uses them for case-based reasoning classification. No other methods do this. ProtoPNet does not reveal concepts, only images. It has 800+ citations and none of them do this.
> > >
> > > The reason this is important is that people don't understand a concept from a single image. **Multiple examples** make it clear (see Figure 1, where we can tell whether it's the shape of the beak, color, image saturation, etc., that is being used because there are multiple images).
> > >
> > > The discussions with other reviewers are not relevant here. That has been a clarification of posthoc methods and heuristic methods vs. non-posthoc methods, and our method is not in the same genre of methods as the posthoc methods they mentioned, which we clarified (and one of their suggestions is not possible at all). Feel free to read our responses to those reviewers.
> > >
> > > >The report lacks specifics regarding the human study—like the number of participants and their expertise (whether they were experts or laypeople), etc…
> > > Your comparison with the random choice (50$\%$) and ProtoPNet—a model with nearly random choice accuracy—doesn’t adequately capture human utility due to the weak baselines employed.
> > >
> > > **A**: Please see the global rebuttal that we had submitted. There were 50 participants, one was thrown away due to sanity checks. The participants rated their expertise with AI.
> > >
> > > That 50\% value reflects that this was an extremely difficult experiment! The HIVE paper had 4 classes, so random guessing would be 25\%, but ProtoPNet got $>50\%$. Thus, the ProtoPNet baseline is *not* weak, as you state.
> > >
> > > In our human studies experiment, we took only the *top two* classes, which makes the problem extremely difficult for humans, so it is not a surprise to get 50\% accuracy. The fact that our method got higher than that is thus very meaningful.
> > >
> > > To add a bit more information on this human studies experiment, the ProtoPNet results were generally lower than our ProtoConcept results -- the median for ProtoPNet on the distinction task was 55\% whereas ProtoConcept's median was 66\%. (There were a few people who got low scores, which reduced the mean).
> > >
> > > We hope this clarification is helpful. We went to a lot of trouble to conduct this experiment quickly to satisfy your request, and the results were quite good. We're not sure what else we could have done...
> > >
> > > >Given that, my primary concerns are still about the novelty and evaluation methodology of this paper. I raise the score (to 4) and am open for more discussion.
> > >
> > > **A**: We hope our clarification about novelty and usefulness, and about the fact that ProtoPNet is not a weak competitor, that we did provide experimental details, and that our results were good will be helpful. Thank you for engaging with us!

---

> > > > ### Comment · Reviewer_p3JV · 2023-08-16
> > > > **Thank you for clarification and final rating**
> > > >
> > > > I am satisfied with the clarification of novelty and evaluation from the authors and decided to raise the score to 6. Thank you for your patience!

---

### Official Review · Reviewer_8ZAP · 2023-07-25

**Soundness:** 3 good
**Presentation:** 3 good
**Contribution:** 3 good
**Rating:** 7
**Confidence:** 3

**Summary:**

This paper improves interpretability for prototypical learning. Most previous methods use a single prototype as an interpretation -- and this is not as informative as we don't know what portion of the prototype corresponds to our target image.

The authors introduce a *prototypical ball* to interpret what prototypes are used for inference. Instead of selecting a single prototype, training samples from this prototypical ball are sampled.

**Strengths:**

*Originality*: I am not completely informed on all prototypical learning methods, but it appears that this method is original. In addition to the novel use of a ball for multiple visualizations, they describe a training scheme specifically for their method. This contributes to the originality of their work.

*Quality*/*Clarity*: The paper is generally presented well, with descriptions of previous work and their drawbacks, as well as what contributions improve the current work. I think some of the descriptions of previous work is not very clear, but I understand space constraints preclude a better description. I believe that the main contribution of this paper can be more clearly elucidated in the figures, which I describe in ``Weaknesses" section.

*Significance*: I think this work is important to the community. While it matches performance of previous methods, it presents a new way of interpreting the results of algorithms. Integrating this novelty into previous works is adds to its significance.


**Weaknesses:**

* A better explanation of interpretability would stregthen the paper. The results do improve interpretability, but the fundamental question (that is introduced in the paper) still remains: why are these selected as prototypes? There are more images and we humans can make connections and inferences, but it's still unclear as to why these are the prototypes within the ball.  I do see this explicitly addressed in Section 5, but I still think it is a weakness of the paper. I think an analysis like your reference [16] would strengthen your paper signficantly. I don't know if it is critical to this paper, but it would be a good follow-up paper.


* Figure 2 is fairly unclear. It's difficult to understand what your method is actually doing. It's not clear how the logits are obtained either. Overall, if I were to look at that diagram without the help of the text, it would be quite unclear as to what your novelty is and how it is implemented. The ball that you introduce, which is your main contribution, is missing from the figure.


A couple small things:

* Please add reference for ProtoPool in Figure 1 and line 209.
* Reference [16] and [18] look very similar -- are these the same?
* The text "ProtoConcepts Layer $g_p$" in Figure 2 looks compressed compared to the rest of the text in that figure.

**Questions:**

* Do you have an ablation on the loss scaling parameters in the supplementary (Tables 2,4)? How were these $\lambda$ chosen? How does performance vary with this?

* Why was $k=10$ chosen for $\mathcal{L}_{\text{clstk}}$? Do you have an ablation on this?

* Continuing from the section above, I think a figure showing the ball, it's center, and radius with its prototypes would help readers understand this work more clearly. For example in Figure 1, the multiple visualizations of prototypical concepts doesn't clearly identify how those multiple visualizations are obtained (i.e. you could use the same figure if the results were the same using a k-NN algorithm) (same for figure 2)

* In equation three, I think it's important that you show the full loss equation i.e. $\mathcal{L} = \mathcal{L}_{\text{other terms}} + \lambda_1 \mathcal{L}_{\text{clstk}} + \lambda_2 \mathcal{L}_{rad} $. It would be clear as to what the loss weights mean in Section 4.1.1


**Limitations:**

The authors have adequately addressed the negative societal impacts.

---

> ### Author Rebuttal · Authors · 2023-08-10
>
> Thank you for your review comments. We are happy to make some clarifications as follows:
> > Why are these selected as prototypes?
>
> **A**: These prototypes are learned. The **data** chooses these prototypes. The interpretation of the prototypes as a concept is done by the human. The algorithm doesn't know what (e.g.) "long beak'' is - it's supposed to discover a concept as a bunch of images of beaks. If you had only one example of a beak, it would just be a picture of a bird's head, and you wouldn't necessarily know what aspect of the beak was important (color? length? shape? combination of those?), but if you had several examples, you could detect the learned concept. Moreover, we performed post-hoc-analysis similar to [1] on the ProtoPool-Concepts examples, and the analysis can be found in the global rebuttal Figure 1. It is worth noting that the texture and shape, as shown in the result, are the two most important factors in learning the "long beak" concept for the given test image of Brown Thrasher. (Note that the ability to learn the prototypes from data is not a weakness; it is a strength since it would find concepts that may not be defined yet by a human. It also makes things a lot easier to assemble the data since the concepts do not need to be defined in advance.)
> >Figure 2 is fairly unclear. It's difficult to understand what your method is actually doing. It's not clear how the logits are obtained either. Overall, if I were to look at that diagram without the help of the text, it would be quite unclear as to what your novelty is and how it is implemented. The ball that you introduce, which is your main contribution, is missing from the figure.
>
> **A**: Thank you for your comment. Because we present a general method which can be added to a wide range of prototype-based models, each of which has a different architecture for calculating logits, we tried to create a figure which describes the prototype-based reasoning process in generality. However, we agree that it can be difficult to understand how the architecture works for a specific model given only the information in Figure 2. Therefore we will create separate architecture figures for each prototype-based model in our experiments and add them to the supplement, so readers can understand the prototype-based reasoning process for each model.
>
> >Do you have an ablation on the loss scaling parameters in the supplementary (Tables 2,4)? How were these chosen? How does performance vary with this?
>
> **A**: The parameters from Tables 2 and 4 are directly borrowed from the previous corresponding model, ProtopNet, and ProtoPool. These are obtained through finetuning by the previous works. For TesNet parameters and the weight of radius loss, we fine-tuned the weight to accommodate our method to the model by grid search. An ablation study on different choices of radius can be found in the Main paper Table 2.
>
> >Why was K=10 chosen for clstk? Do you have an ablation on this?
>
>  **A**: K is a hyperparameter of our model and is selected through parameter tuning by grid search. The ablation of different choices of Ks on the ProtoPool-based model can be found in Supplementary material Table 1 .
>
> >A couple small things:
> Please add reference for ProtoPool in Figure 1 and line 209.
> Reference [16] and [18] look very similar -- are these the same?
> The text "ProtoConcepts Layer
> " in Figure 2 looks compressed compared to the rest of the text in that figure.
>
> **A**:Thank you for catching these small errors. We will fix these issues in our final manuscript.
>
> [1] Meike Nauta, Annemarie Jutte, Jesper Provoost, and Christin Seifert. This looks like that, because ... explaining prototypes for interpretable image recognition. In Machine Learning and Principles and Practice of Knowledge Discovery in Databases, pages 441–456. Springer International Publishing, 2021. ISBN 978-3-030-93736-2.

---

> > ### Comment · Reviewer_8ZAP · 2023-08-13
> >
> > Than you for your response.
> >
> > *A: Thank you for your comment. Because we present a general method which can be added to a wide range of prototype-based models, each of which has a different architecture for calculating logits, we tried to create a figure which describes the prototype-based reasoning process in generality. However, we agree that it can be difficult to understand how the architecture works for a specific model given only the information in Figure 2. Therefore we will create separate architecture figures for each prototype-based model in our experiments and add them to the supplement, so readers can understand the prototype-based reasoning process for each model.*
> >
> > I think what I mean is: the core algorithmic idea (as I understand it) is generating the prototypical ball from which prototypes are generated. It would be good to see this in the figure. I'm not sure if an architecture-specific figure is needed, but I would like to see your core contribution.
> >
> > You have sufficiently addressed my concerns. I think the global rebuttal and results presented there improve the strength of the paper. If the authors commit to including the improvements listed in their individual and global rebuttals, I will increase my score "Accept"

---

> > > ### Author Response · Authors · 2023-08-14
> > >
> > > Thank you for your comments and clarifications!
> > >
> > > We will add our core algorithmic idea (prototypical ball) element in Figure 2 for the final manuscript. And we will add the survey and its result to the final manuscript.

---

### Author Rebuttal · Authors · 2023-08-10

**User Study** We thank all the reviewers for their comments. To show the reduction of ambiguity (and resulting improvement in user interpretability), we created a **distinction** user study similar to HIVE[1] to compare our ProtoPConcepts method with ProtoPNet. We randomly picked ten samples from the test set and calculated the top two predicted classes (i.e., the classes with the highest predicted probabilities according to the model) for each test sample. We then provided visual explanations from the most activated prototypes for these classes by ProtoPNet and ProtoPNet-Concepts. A test-taker was then asked to choose which class the model is actually predicting, looking only at the visual explanations without the class probabilities. Examples of our user study are shown in the global rebuttal pdf Figure 2. We released our user study on Amazon Mechanical Turk and collected 50 responses from test takers with a 98% survey approval rate  to ensure the quality of responses, and removed 1 response from both surveys after screening for nonsensical free response answers. We first ran a two-sided t-test on self-rated ML experience for the test takers from the ProtoPNet and ProtoPNet-Concept. The p-value is 1, and we are assured that there is no statistically significant difference in machine learning experience between the two groups on average. We further performed a one-sided, two-sample Welch t-test with the alternative hypothesis that the ProtoConcept user study results in higher accuracy than the ProtoPNet on average. The p-value is 0.003, which means that **ProtoConcept exhibited a statistically significant improvement in model interpretability over ProtoPNet**. Moreover, users given visual explanations from ProtoPNet could not statistically beat a random guessing accuracy of 50 percent (p=0.289), which is consistent with the findings in the HIVE paper[1]. With our model, users' mean accuracy was able to beat random guessing by a statistically significant margin (p=$2.85*e^{-5}$). Our survey results show not only that **our model provides a notable improvement in user interpretability**, but is able to **improve non-expert user performance** in a difficult fine-grained classification task whereas the previous ProtoPNet model cannot. The detailed test statistics can be found in global rebuttal pdf Table 1.

[1] Kim et al., HIVE: evaluating the human interpretability of visual explanations, ECCV 2022

---

### Decision · Program_Chairs · 2023-09-21

**Decision:**

Accept (poster)

**Comment:**

Summary
This paper improves upon prototype-based learning for image classification. While most prior work use a one-to-one mapping, this paper proposes a way to use multiple training image patches per prototype. This makes the  visualization of the prototypes more interpretable. The method is evaluated on image classification using the CUB and the Cars dataset.

Reviews & Justification
The paper received positive reviews overall and the reviewers were satisfied with the author responses. The reviewers find this paper provides a significant improvement over prior variants of the ProtoPNets.